# LEARNABLE UNCERTAINTY UNDER LAPLACE APPROXIMATIONS

## ABSTRACT

Laplace approximations are classic, computationally lightweight means for constructing Bayesian neural networks (BNNs). As in other approximate BNNs, one cannot necessarily expect the induced predictive uncertainty to be calibrated. Here we develop a formalism to explicitly "train" the uncertainty in a decoupled way to the prediction itself. To this end we introduce *uncertainty units* for Laplace-approximated networks: Hidden units with zero weights that can be added to any pre-trained, point-estimated network. Since these units are inactive, they do not affect the predictions. But their presence changes the geometry (in particular the Hessian) of the loss landscape around the point estimate, thereby affecting the network's uncertainty estimates under a Laplace approximation. We show that such units can be trained via an uncertainty-aware objective, making the Laplace approximation competitive with more expensive alternative uncertainty-quantification frameworks.

## 1 INTRODUCTION

The point estimates of neural networks (NNs)—constructed as *maximum a posteriori* (MAP) estimates via (regularized) empirical risk minimization—empirically achieve high predictive performance. However, they tend to underestimate the uncertainty of their predictions, leading to an overconfidence problem (Hein et al., 2019), which could be disastrous in safety-critical applications such as autonomous driving. Bayesian inference offers a principled path to overcome this issue. The goal is to turn a "vanilla" NN into a Bayesian neural network (BNN), where the posterior distribution over the network's weights are inferred via Bayes' rule and subsequently taken into account when making predictions. Since the cost of exact posterior inference in a BNN is often prohibitive, approximate Bayesian methods are employed instead.

Laplace approximations (LAs) are classic methods for such a purpose (MacKay, 1992b). The key idea is to obtain an approximate posterior by "surrounding" a MAP estimate of a network with a Gaussian, based on the loss landscape's geometry around it. A standard practice in LAs is to tune a single hyperparameter—the prior precision—which is inflexible (Ritter et al., 2018b; Kristiadi et al., 2020). Here, we aim at improving the flexibility of uncertainty tuning in LAs. To this end, we introduce *Learnable Uncertainty under Laplace Approximations (LULA) units*, which are hidden units associated with a zeroed weight. They can be added to the hidden layers of any MAP-trained network. Because they are inactive, such units do not affect the prediction of the underlying network. However, they can still contribute to the Hessian of the loss with respect to the parameters, and hence induce additional structures to the posterior covariance under a LA. LULA units can be trained via an uncertainty-aware objective (Hendrycks et al., 2019; Hein et al., 2019, etc.), such that they improve the predictive uncertainty-quantification (UQ) performance of the Laplace-approximated BNN. Figure 1 demonstrates trained LULA units in action: They improve the UQ performance of a standard LA, while keeping the MAP predictions in both regression and classification tasks.

In summary, we (i) introduce LULA units: inactive hidden units for uncertainty tuning of a LA, (ii) bring a robust training technique from non-Bayesian literature for training these units, and (iii) show empirically that LULA-augmented Laplace-approximated BNNs can yield better UQ performance compared to both previous tuning techniques and contemporary, more expensive baselines.

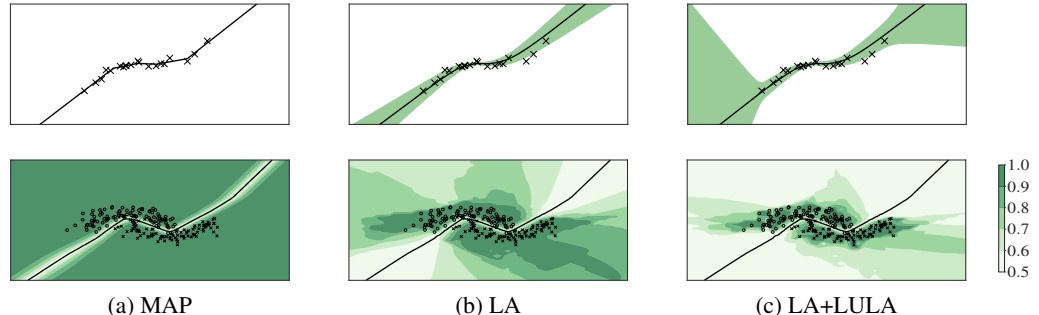

(a) MAP                    (b) LA                    (c) LA+LULA

Figure 1: Predictive uncertainty of a **(a)** MAP-trained, **(b)** Laplace-approximated (LA), and **(c)** LULA-augmented Laplace-approximated (LA+LULA) neural network on regression **(top)** and classification **(bottom)** tasks. Black curve represent predictive mean and decision boundary, and shade represents $\pm 3$ standard deviation and confidence in regression and classification, respectively. MAP is overconfident and LA can mitigate this. However, LA can still be overconfident away from the data. LULA improve LA's uncertainty further without affecting its predictions.

## 2 BACKGROUND

### 2.1 BAYESIAN NEURAL NETWORKS

Let $f : \mathbb{R}^n \times \mathbb{R}^d \to \mathbb{R}^k$ defined by $(x, \theta) \mapsto f(x; \theta)$ be an $L$-layer neural network. Here, $\theta$ is the concatenation of all the parameters of $f$. Suppose that the size of each layer of $f$ is given by the sequence of $(n_l \in \mathbb{Z}_{>0})_{l=1}^L$. Then, for each $l = 1, \ldots, L$, the $l$-th layer of $f$ is defined by

$$a^{(l)} := W^{(l)} h^{(l-1)} + b^{(l)}, \qquad \text{with} \quad h^{(l)} := \begin{cases} \varphi(a^{(l)}) & \text{if } l < L \\ a^{(l)} & \text{if } l = L, \end{cases} \tag{1}$$

where $W^{(l)} \in \mathbb{R}^{n_l \times n_{l-1}}$ and $b^{(l)} \in \mathbb{R}^{n_l}$ are the weight matrix and bias vector of the layer, and $\varphi$ is a component-wise activation function. We call the vector $h^{(l)} \in \mathbb{R}^{n_l}$ the $l$-th hidden units of $f$. Note that by convention, we consider $n_0 := n$ and $n_L := k$, while $h^{(0)} := x$ and $h^{(L)} := f(x; \theta)$.

From the Bayesian perspective, the ubiquitous training formalism of neural networks amounts to MAP estimation: The empirical risk and the regularizer are interpretable as the negative log-likelihood under an i.i.d. dataset $\mathcal{D} := \{x_i, y_i\}_{i=1}^m$ and the negative log-prior, respectively. That is, the loss function is interpreted as

$$\mathcal{L}(\theta) := -\sum_{i=1}^m \log p(y_i \mid f(x_i; \theta)) - \log p(\theta) = -\log p(\theta \mid \mathcal{D}). \tag{2}$$

In this view, the *de facto* weight decay regularizer amounts to a zero-mean isotropic Gaussian prior $p(\theta) = \mathcal{N}(\theta \mid 0, \lambda^{-1} I)$ with a scalar precision hyperparameter $\lambda$. Meanwhile, the usual softmax and quadratic output losses correspond to the Categorical and Gaussian distributions over $y_i$ in the case of classification and regression, respectively.

MAP-trained neural networks have been shown to be overconfident (Hein et al., 2019) and BNNs can mitigate this issue (Kristiadi et al., 2020). They quantify epistemic uncertainty by inferring the full posterior distribution of the parameters $\theta$ (instead of just a single point estimate in MAP training). Given that $p(\theta \mid \mathcal{D})$ is the posterior, then the prediction for any test point $x \in \mathbb{R}^n$ is obtained via marginalization

$$p(y \mid x, \mathcal{D}) = \int p(y \mid f(x; \theta)) \, p(\theta \mid \mathcal{D}) \, d\theta, \tag{3}$$

which captures the uncertainty encoded in the posterior.

## 2.2 LAPLACE APPROXIMATIONS

In deep learning, since the exact Bayesian posterior is intractable, approximate Bayesian inference methods are used. An important family of such methods is formed by LAs. Let $\theta_{\mathrm{MAP}}$ be the minimizer of (2), which corresponds to a mode of the posterior distribution. A LA locally approximates the posterior using a Gaussian

$$p(\theta \mid \mathcal{D}) \approx \mathcal{N}(\theta \mid \theta_{\mathrm{MAP}}, \Sigma) := \mathcal{N}(\theta \mid \theta_{\mathrm{MAP}}, (\nabla^2 \mathcal{L}|_{\theta_{\mathrm{MAP}}})^{-1}) \,.$$

Thus, LAs construct an approximate Gaussian posterior *around* $\theta_{\mathrm{MAP}}$, whose precision equals the Hessian of the loss at $\theta_{\mathrm{MAP}}$—the "curvature" of the loss landscape at $\theta_{\mathrm{MAP}}$. While the covariance of a LA is tied to the weight decay of the loss, a common practice in LAs is to tune the prior precision under some objective, in a *post-hoc* manner. In other words, the MAP estimation and the covariance inference are thought as separate, independent processes. For example, given a fixed MAP estimate, one can maximize the log-likelihood of a LA w.r.t. the prior precision to obtain the covariance. This hyperparameter tuning can thus be thought as an *uncertainty tuning*.

A recent example of LAs is the Kronecker-factored Laplace (KFL, Ritter et al., 2018b). The key idea is to approximate the Hessian matrix with the layer-wise Kronecker factorization scheme proposed by Heskes (2000); Martens & Grosse (2015). That is, for each layer $l = 1, \dots, L$, KFL assumes that the Hessian corresponding to the $l$-th weight matrix $W^{(l)} \in \mathbb{R}^{n_l \times n_{l-1}}$ can be written as the Kronecker product $G^{(l)} \otimes A^{(l)}$ for some $G^{(l)} \in \mathbb{R}^{n_l \times n_l}$ and $A^{(l)} \in \mathbb{R}^{n_{l-1} \times n_{l-1}}$. This assumption brings the inversion cost of the Hessian down to $\Theta(n_l^3 + n_{l-1}^3)$, instead of the usual $\Theta(n_l^3 n_{l-1}^3)$ cost. The approximate Hessian can easily be computed via tools such as BackPACK (Dangel et al., 2020).

Even with a closed-form Laplace-approximated posterior, the predictive distribution (3) in general does not have an analytic solution since $f$ is nonlinear. Instead, one can employ Monte-Carlo (MC) integration by sampling from the Gaussian:

$$p(y \mid x, \mathcal{D}) \approx \frac{1}{S} \sum_{s=1}^{S} p(y \mid f(x; \theta_s)); \qquad \theta_s \sim \mathcal{N}(\theta \mid \theta_{\mathrm{MAP}}, \Sigma) \,,$$

for $S$ number of samples. In the case of binary classification with $f : \mathbb{R}^n \times \mathbb{R}^d \to \mathbb{R}$, one can use the following well-known approximation, due to MacKay (1992a):

$$p(y = 1 \mid x, \mathcal{D}) \approx \sigma\left( \frac{f(x; \theta_{\mathrm{MAP}})}{\sqrt{1 + \pi/8 \, v(x)}} \right) , \tag{4}$$

where $\sigma$ is the logistic-sigmoid function and $v(x)$ is the marginal variance of the network output $f(x)$, which is often approximated via a linearization of the network around the MAP estimate:

$$v(x) \approx (\nabla_\theta f(x; \theta)|_{\theta_{\mathrm{MAP}}})^\top \Sigma \, (\nabla_\theta f(x; \theta)|_{\theta_{\mathrm{MAP}}}) \,. \tag{5}$$

(This approximation has also been generalized to multi-class classifications by Gibbs (1997).) In particular, as $v(x)$ increases, the predictive probability of $y = 1$ goes to 0.5 and therefore the uncertainty increases. This relationship has also been shown empirically in multi-class classifications with MC-integration (Kristiadi et al., 2020).

## 3 LULA UNITS

The problem with the standard uncertainty tuning in LAs is that the only degree-of-freedom available for performing the optimization is the *scalar* prior precision and therefore inflexible.[1] We shall address this by introducing "uncertainty units", which can be added on top of the hidden units of any MAP-trained network (Section 3.1) and can be trained via an uncertainty-aware loss (Section 3.2).

### 3.1 CONSTRUCTION

Let $f : \mathbb{R}^n \times \mathbb{R}^d \to \mathbb{R}^k$ be a MAP-trained $L$-layer neural network with parameters $\theta_{\mathrm{MAP}} = \{W_{\mathrm{MAP}}^{(l)}, b_{\mathrm{MAP}}^{(l)}\}_{l=1}^{L}$. The premise of our method is simple: At each hidden layer $l = 1, \dots, L-1$,

---

[1]While one can also use a non-scalar prior precision, it appears to be uncommon in deep learning. In any case, such a element-wise weight-cost would interact with the training procedure.

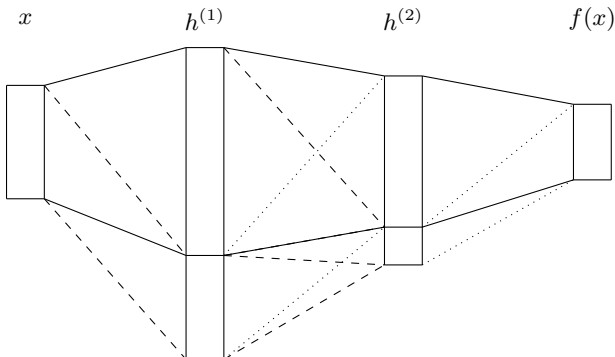

Figure 2: An illustration of the proposed construction. **Rectangles** represent layers, **solid lines** represent connection between layers, given by the original weight matrices $W_{\mathrm{MAP}}^{(1)}, \ldots, W_{\mathrm{MAP}}^{(L)}$. The additional units are represented by the additional block at the bottom of each layer. **Dashed lines** correspond to the free parameters $\widehat{W}^{(1)}, \ldots, \widehat{W}^{(L-1)}$, while **dotted lines** to the zero weights.

suppose we add $m_l \in \mathbb{Z}_{\geq 0}$ additional hidden units, under the original activation function, to $h^{(l)}$. As a consequence, we need to augment each of the weight matrices to accommodate them.

Consider the following construction: for each layer $l = 1, \ldots, L - 1$ of the network, we expand $W^{(l)}$ and $b^{(l)}$ to obtain the block matrix and vector

$$\widetilde{W}^{(l)} := \begin{pmatrix} W_{\mathrm{MAP}}^{(l)} & 0 \\ \widehat{W}_1^{(l)} & \widehat{W}_2^{(l)} \end{pmatrix} \in \mathbb{R}^{(n_l+m_l)\times(n_{l-1}+m_{l-1})}; \qquad \widetilde{b}^{(l)} := \begin{pmatrix} b_{\mathrm{MAP}}^{(l)} \\ \widehat{b}^{(l)} \end{pmatrix} \in \mathbb{R}^{n_l+m_l}, \qquad (6)$$

respectively, with $m_0 = 0$ since we do not add additional units to the input. For $l = L$, we define

$$\widetilde{W}^{(L)} := (W_{\mathrm{MAP}}^{(L)}, 0) \in \mathbb{R}^{k \times (n_{L-1}+m_{L-1})}; \qquad \widetilde{b}^{(L)} := b_{\mathrm{MAP}}^{(L)} \in \mathbb{R}^k,$$

so that the output dimensionality is unchanged. For brevity, we denote $\widehat{W}^{(l)} := (\widehat{W}_1^{(l)}, \widehat{W}_2^{(l)})$. Refer to Figure 2 for an illustration and Algorithm 2 in Appendix B for a step-by-step summary. Taken together, we denote the resulting augmented network as $\widetilde{f}$ and the resulting parameter vector as $\widetilde{\theta}_{\mathrm{MAP}} \in \mathbb{R}^{\widetilde{d}}$, where $\widetilde{d}$ it the resulting number of parameters. Note that we can easily extend this construction to convolutional nets by expanding the "channel" of a hidden layer.[2]

Let us inspect the implication of this construction. Here for each $l = 1, \ldots, L - 1$, since they are zero, the upper-right quadrant of $\widetilde{W}^{(l)}$ *deactivates* the $m_{l-1}$ additional hidden units in the previous layer, thus they do not contribute to the original hidden units in the $l$-th layer. Meanwhile, the sub-matrix $\widehat{W}^{(l)}$ and the sub-vector $\widehat{b}^{(l)}$ contain parameters for the additional $m_l$ hidden units in the $l$-th layer. We are free to choose the the values of these parameters since the following proposition guarantees that they will not change the output of the network (the proof is in Appendix A).

**Proposition 1.** *Let $f : \mathbb{R}^n \times \mathbb{R}^d \to \mathbb{R}^k$ be a MAP-trained $L$-layer network parametrized by $\theta_{\mathrm{MAP}}$. Suppose $\widetilde{f} : \mathbb{R}^n \times \mathbb{R}^{\widetilde{d}} \to \mathbb{R}$ and $\widetilde{\theta}_{\mathrm{MAP}} \in \mathbb{R}^{\widetilde{d}}$ are obtained via the previous construction. For any input $x \in \mathbb{R}^n$, we have $\widetilde{f}(x; \widetilde{\theta}_{\mathrm{MAP}}) = f(x; \theta_{\mathrm{MAP}})$.*

So far, it looks like all our changes to the network are inconsequential. However, they do affect the *curvature* of the landscape of $\mathcal{L}$,[3] and thus the uncertainty arising in a LA. Let $\widetilde{\theta}$ be a random variable in $\mathbb{R}^{\widetilde{d}}$ and $\widetilde{\theta}_{\mathrm{MAP}}$ be an instance of it. Suppose we have a Laplace-approximated posterior $p(\widetilde{\theta} \mid \mathcal{D}) \approx \mathcal{N}(\widetilde{\theta} \mid \widetilde{\theta}_{\mathrm{MAP}}, \widetilde{\Sigma})$ over $\widetilde{\theta}$, where the covariance $\widetilde{\Sigma}$ is the inverse Hessian of the negative log-posterior w.r.t. the augmented parameters at $\widetilde{\theta}_{\mathrm{MAP}}$. Then, $\widetilde{\Sigma}$ contains additional dimensions (and thus in general, additional structured, non-zero uncertainty) absent in the original network, which depend on the values of the free parameters $\{\widehat{W}^{(l)}, \widehat{b}^{(l)}\}_{l=1}^{L-1}$.

---

[2]E.g. if the hidden units are a 3D array of (channel $\times$ height $\times$ width), then we expand the first dimension.
[3]More formally: The principal curvatures of the graph of $\mathcal{L}$, seen as a $d$-dimensional submanifold of $\mathbb{R}^{d+1}$.

The implication of the previous finding can be seen clearly in real-valued networks with diagonal LA posteriors. The following proposition shows that, under such a network and posterior, the construction above will affect the *output uncertainty* of the original network $f$ (the proof is in Appendix A).

**Proposition 2.** *Suppose $f : \mathbb{R}^n \times \mathbb{R}^d \to \mathbb{R}$ is a real-valued network and $\widetilde{f}$ is as constructed above. Suppose further that diagonal Laplace-approximated posteriors $\mathcal{N}(\theta \mid \theta_{\mathrm{MAP}}, \mathrm{diag}(\sigma))$, $\mathcal{N}(\widetilde{\theta} \mid \widetilde{\theta}, \mathrm{diag}(\widetilde{\sigma}))$ are employed. Using the linearization (5), for any input $x \in \mathbb{R}^n$, the variance over the output $\widetilde{f}(x; \widetilde{\theta})$ is at least that of $f(x; \theta)$.*

In summary, the construction along with Propositions 1 and 2 imply that the additional hidden units we have added to the original network are *uncertainty units* under LAs, i.e. hidden units that *only* contribute to the Laplace-approximated uncertainty and not the predictions. This property gives rise to the name *Learnable Uncertainty under Laplace Approximations (LULA) units*.

## 3.2 TRAINING

We have seen that by adding LULA units to a network, we obtain additional free parameters that only affect uncertainty under a LA. These parameters are thus useful for uncertainty calibration. Our goal is therefore to train them to induce low uncertainty over the data (inliers) and high uncertainty on outliers—the so-called out-of-distribution (OOD) data. Specifically, this can be done by minimizing the *output variance* over inliers while maximizing it over outliers. Note that using variance makes sense in both the regression and classification cases: In the former, this objective directly maintains narrow error bars near the data while widen those far-away from them—cf. Figure 1 (c, top). Meanwhile, in classifications, variances over function outputs directly impact predictive confidences, as we have noted in the discussion of (4)—higher variance implies lower confidence.

Thus, following the contemporary technique from non-Bayesian robust learning literature (Hendrycks et al., 2019; Hein et al., 2019, etc.), we construct the following loss. Let $f : \mathbb{R}^n \times \mathbb{R}^d \to \mathbb{R}^k$ be an $L$-layer neural network with a MAP-trained parameters $\theta_{\mathrm{MAP}}$ and let $\widetilde{f} : \mathbb{R}^n \times \mathbb{R}^{\widetilde{d}} \to \mathbb{R}^k$ along with $\widetilde{\theta}_{\mathrm{MAP}}$ be obtained by adding LULA units. Denoting the dataset sampled i.i.d. from the data distribution as $\mathcal{D}_{\mathrm{in}}$ and that from some outlier distribution as $\mathcal{D}_{\mathrm{out}}$, we define

$$\mathcal{L}_{\mathrm{LULA}}(\widetilde{\theta}_{\mathrm{MAP}}) := \frac{1}{|\mathcal{D}_{\mathrm{in}}|} \sum_{x_{\mathrm{in}} \in \mathcal{D}_{\mathrm{in}}} \nu(\widetilde{f}(x_{\mathrm{in}}); \widetilde{\theta}_{\mathrm{MAP}}) - \frac{1}{|\mathcal{D}_{\mathrm{out}}|} \sum_{x_{\mathrm{out}} \in \mathcal{D}_{\mathrm{out}}} \nu(\widetilde{f}(x_{\mathrm{out}}); \widetilde{\theta}_{\mathrm{MAP}}), \tag{7}$$

where $\nu(\widetilde{f}(x); \widetilde{\theta}_{\mathrm{MAP}})$ is the total variance over the $k$ components of the network output $\widetilde{f}_1(x; \widetilde{\theta}), \ldots, \widetilde{f}_k(x; \widetilde{\theta})$ under the Laplace-approximated posterior $p(\widetilde{\theta} \mid \mathcal{D}) \approx \mathcal{N}(\widetilde{\theta} \mid \widetilde{\theta}_{\mathrm{MAP}}, \widetilde{\Sigma}(\widetilde{\theta}_{\mathrm{MAP}}))$, which can be approximated via an $S$-samples MC-integral

$$
\begin{aligned}
\nu(\widetilde{f}(x); \widetilde{\theta}_{\mathrm{MAP}}) &:= \sum_{i=1}^{k} \mathrm{var}_{p(\widetilde{\theta} \mid \mathcal{D})} \widetilde{f}_i(x; \widetilde{\theta}) \\
&\approx \sum_{i=1}^{k} \left( \frac{1}{S} \sum_{s=1}^{S} \widetilde{f}_i(x; \widetilde{\theta}_s)^2 \right) - \left( \frac{1}{S} \sum_{s=1}^{S} \widetilde{f}_i(x; \widetilde{\theta}_s) \right)^2 ; \quad \text{with } \widetilde{\theta}_s \sim p(\widetilde{\theta} \mid \mathcal{D}).
\end{aligned}
\tag{8}
$$

Here, for clarity, we have shown explicitly the dependency of $\widetilde{\Sigma}$ on $\widetilde{\theta}_{\mathrm{MAP}}$. Note that we can simply set $\mathcal{D}_{\mathrm{in}}$ to be the training set of a dataset. Furthermore, throughout this paper, we use the simple OOD dataset proposed by Hein et al. (2019) which is constructed via permutation, blurring, and contrast rescaling of the in-distribution dataset $\mathcal{D}_{\mathrm{in}}$. As we shall show in Section 5, this artificial, uninformative OOD dataset is sufficient for obtaining good results across benchmark problems. More complex dataset as $\mathcal{D}_{\mathrm{out}}$ might improve LULA's performance further but is not strictly necessary.

Since our aim is to solely improve the uncertainty, we must maintain the structure of all weights and biases in $\widetilde{\theta}_{\mathrm{MAP}}$, in accordance to (6). This can simply be enforced via gradient masking: For all $l = 1, \ldots, L-1$, set the gradients of the blocks of $\widetilde{W}^{(l)}$ and $\widetilde{b}^{(l)}$ not corresponding to $\widehat{W}^{(l)}$ and $\widehat{b}^{(l)}$, respectively, to zero. Furthermore, since the covariance matrix $\widetilde{\Sigma}(\widetilde{\theta})$ of the Laplace-approximated posterior is a function of $\widetilde{\theta}$, it needs to be updated at every iteration during the optimization of $\mathcal{L}_{\mathrm{LULA}}$.

---

**Algorithm 1** Training LULA units.

**Input:**
    MAP-trained network $f$. Dataset $\mathcal{D}$, OOD dataset $\mathcal{D}_{\text{out}}$. Learning rate $\alpha$. Number of epochs $E$.
1: Construct $\widetilde{f}$ from $f$ by following Section 3.1.
2: **for** $i = 1, \ldots, E$ **do**
3:      $p(\widetilde{\theta} \mid \mathcal{D}) \approx \mathcal{N}(\widetilde{\theta} \mid \widetilde{\theta}_{\text{MAP}}, \widetilde{\Sigma}(\widetilde{\theta}_{\text{MAP}}))$          ▷ Obtain a Laplace-approximated posterior of $\widetilde{f}$
4:      Compute $\mathcal{L}_{\text{LULA}}(\widetilde{\theta}_{\text{MAP}})$ via (7) using $p(\widetilde{\theta} \mid \mathcal{D})$, $\mathcal{D}$, and $\mathcal{D}_{\text{out}}$
5:      $g = \nabla \mathcal{L}_{\text{LULA}}(\widetilde{\theta}_{\text{MAP}})$
6:      $\widehat{g} = \texttt{mask\_gradient}(g)$          ▷ Zero out the derivatives *not* corresponding to $\widehat{\theta}$
7:      $\widetilde{\theta}_{\text{MAP}} = \widetilde{\theta}_{\text{MAP}} - \alpha \widehat{g}$
8: **end for**
9: $p(\widetilde{\theta} \mid \mathcal{D}) \approx \mathcal{N}(\widetilde{\theta} \mid \widetilde{\theta}_{\text{MAP}}, \widetilde{\Sigma}(\widetilde{\theta}_{\text{MAP}}))$          ▷ Obtain the final Laplace approximation
10: **return** $\widetilde{f}$ and $p(\widetilde{\theta} \mid \mathcal{D})$

---

The cost scales in the network's depth and can thus be expensive. Inspired by recent findings that last-layer Bayesian methods are competitive to all-layer alternatives (Ober & Rasmussen, 2019), we thus consider a last-layer LA (Kristiadi et al., 2020) as a proxy: We apply a LA only at the last hidden layer and assume that the first $L - 2$ layers are learnable feature extractor. We use a diagonal last-layer Fisher matrix to approximate the last-layer Hessian. Note that backpropagation through this matrix does not pose a difficulty since modern deep learning libraries such as PyTorch and TensorFlow supports "double backprop" (backpropagation through a gradient) efficiently. Finally, the loss $\mathcal{L}_{\text{LULA}}$ can be minimized using standard gradient-based optimizers. Refer to Algorithm 1 for a summary.

Last but not least, the intuition of LULA training is as follows. By adding LULA units, we obtain an augmented version of the network's loss landscape. The goal of LULA training is then to exploit the weight-space symmetry (i.e. different parameters but induce the same output) arisen from the construction as shown by Proposition 1 and pick one of these parameters that is symmetric to the original parameter but has "better" curvatures. Here, we define a "good curvature" in terms of the above objective. These curvatures, then, when used in a Laplace approximation, could yield better uncertainty estimates compared to the standard non-LULA-augmented Laplace approximations.

## 4    RELATED WORK

While traditionally hyperparameter optimization in a LA requires re-training of the network—under the evidence framework (MacKay, 1992b) or empirical Bayes (Bernardo & Smith, 2009), tuning it in a *post-hoc* manner has increasingly becomes a common practice. Ritter et al. (2018a;b) tune the prior precision of a LA by maximizing the predictive log-likelihood. Kristiadi et al. (2020) extend this procedure by also using OOD data to better calibrate the uncertainty. However, they are limited in terms of flexibility since the prior precision of the LAs constitutes to just a single parameter. LULA can be seen as an extension of these approaches with greater flexibility and is complementary to them since LULA is independent to the prior precision.

Confidence calibration via OOD data has achieved state-of-the-art performance in non-Bayesian outlier detection. Hendrycks et al. (2019); Hein et al. (2019); Meinke & Hein (2020) use OOD data to regularize the standard maximum-likelihood training. Malinin & Gales (2018; 2019) use OOD data to train probabilistic models based on the Dirichlet distribution. All these methods are non-Bayesian and non-*post-hoc*. Our work is thus orthogonal since we aim at improving a class of Bayesian models in a *post-hoc* manner. LULA can be seen as bringing the best of both worlds: Bayesian uncertainty that is tuned via the state-of-the-art non-Bayesian technique.

## 5    EXPERIMENTS

In this section, we focus on classification using the standard OOD benchmark problems. Supplementary results on regression are in Section C.2.

Table 1: Average confidences (MMCs in percent) over ten prediction runs. Lower is better for OOD data while higher is better for in-distribution data. See Table 2 for the AUR values.

| Train - Test | MAP | MCD | DE | KFL | KFL+LL | KFL+OOD | *KFL+LULA* |
|---|---|---|---|---|---|---|---|
| MNIST | 98.0±0.0 | 97.4±0.0 | 98.0±0.0 | 96.0±0.1 | 98.0±0.0 | 96.6±0.0 | 96.6±0.1 |
| E-MNIST | 81.4±0.0 | 76.7±0.0 | 79.7±0.0 | 72.1±0.2 | 81.3±0.0 | 73.7±0.2 | **71.4±0.4** |
| F-MNIST | 76.6±0.0 | 67.8±0.0 | 74.7±0.0 | 65.8±0.4 | 76.5±0.0 | 67.4±0.4 | **49.4±1.9** |
| Uniform Noise | 99.5±0.0 | 97.0±0.0 | 98.8±0.0 | 86.3±1.4 | 99.5±0.0 | 89.8±0.9 | **33.8±2.5** |
| Asymptotic | 100.0±0.0 | 88.0±0.0 | 75.9±0.0 | **45.9±0.9** | 97.3±0.1 | 50.4±1.3 | 46.0±2.6 |
| CIFAR-10 | 95.2±0.0 | 79.0±0.0 | 92.9±0.0 | 91.0±0.2 | 93.4±0.1 | 85.3±0.6 | 88.8±0.3 |
| SVHN | 75.5±0.0 | 62.7±0.0 | 65.7±0.0 | 61.5±0.6 | 68.4±0.5 | 52.3±1.1 | **37.5±1.7** |
| LSUN-CR | 61.2±0.0 | 39.7±0.2 | 48.0±0.0 | 48.1±1.0 | 54.5±0.6 | 40.9±1.0 | **23.1±1.5** |
| CIFAR-100 | 79.3±0.0 | **56.1±0.0** | 69.5±0.0 | 67.3±0.6 | 73.5±0.2 | 57.4±0.8 | 63.2±0.4 |
| Uniform Noise | 51.5±0.0 | 72.4±0.0 | **33.1±0.0** | 43.5±0.6 | 46.5±0.6 | 43.5±1.1 | 34.4±1.9 |
| Asymptotic | 100.0±0.0 | 49.3±0.1 | **41.2±0.0** | 78.9±2.0 | 90.4±1.0 | 61.3±2.0 | 41.4±2.0 |
| SVHN | 97.7±0.0 | 92.7±0.0 | 94.4±0.0 | 95.1±0.1 | 96.4±0.0 | 96.6±0.0 | 91.5±0.5 |
| CIFAR-10 | 64.3±0.0 | 40.8±0.0 | 42.2±0.0 | 48.6±0.3 | 54.7±0.3 | 56.2±0.2 | **40.2±0.6** |
| LSUN-CR | 60.3±0.0 | **33.4±0.1** | 36.4±0.0 | 49.0±0.5 | 54.0±0.4 | 55.0±0.4 | 36.0±1.0 |
| CIFAR-100 | 66.3±0.0 | 43.1±0.0 | 44.7±0.0 | 49.7±0.3 | 56.1±0.3 | 57.7±0.2 | **42.0±0.5** |
| Uniform Noise | 53.6±0.0 | 25.9±0.0 | 29.8±0.0 | 41.5±0.5 | 46.8±0.3 | 47.9±0.3 | **22.9±0.6** |
| Asymptotic | 99.7±0.0 | 73.8±0.1 | 74.2±0.0 | 35.8±0.7 | 49.4±0.6 | 53.7±0.6 | **22.4±0.3** |
| CIFAR-100 | 81.7±0.0 | 57.1±0.0 | 65.0±0.0 | 57.3±0.2 | 70.3±0.1 | 66.4±0.1 | 44.3±0.2 |
| SVHN | 69.9±0.0 | 35.6±0.0 | 43.6±0.0 | 40.2±0.8 | 54.4±0.4 | 49.7±0.7 | **23.4±0.5** |
| LSUN-CR | 60.4±0.0 | 29.8±0.2 | 47.0±0.0 | 39.4±1.5 | 51.0±0.5 | 46.8±0.9 | **14.3±1.0** |
| CIFAR-10 | 71.5±0.0 | 39.2±0.0 | 47.0±0.0 | 42.6±0.2 | 56.8±0.1 | 52.2±0.2 | **31.8±0.2** |
| Uniform Noise | 86.3±0.0 | **27.8±0.0** | 46.8±0.0 | 56.4±3.9 | 74.2±0.8 | 67.6±2.4 | 34.3±2.0 |
| Asymptotic | 99.9±0.0 | 40.7±0.1 | 25.5±0.0 | 21.8±0.9 | 34.0±1.2 | 29.3±0.8 | **12.6±0.2** |

Table 2: OOD detection performance measured by the AUR metric. Values reported are averages over ten prediction runs. Higher is better. Underline and bold faces indicate the highest values over the last four columns and all columns in a given row, respectively.

| Train - Test | MAP | MCD | DE | KFL | KFL+LL | KFL+OOD | *KFL+LULA* |
|---|---|---|---|---|---|---|---|
| *MNIST* | | | | | | | |
| E-MNIST | 84.8±0.0 | 86.1±0.0 | 85.8±0.0 | 86.3±0.1 | 84.8±0.0 | 86.3±0.1 | **88.3±0.2** |
| F-MNIST | 86.8±0.0 | 94.9±0.0 | 89.5±0.0 | 91.3±0.3 | 86.9±0.0 | 91.0±0.3 | **98.0±0.4** |
| Uniform Noise | 62.8±0.0 | 80.0±0.1 | 76.7±0.0 | 90.0±1.0 | 62.9±0.1 | 89.0±0.7 | **99.8±0.1** |
| Asymptotic | 0.2±0.0 | 75.0±0.3 | 67.9±0.0 | **99.0±0.1** | 12.5±0.3 | 98.7±0.2 | **99.0±0.2** |
| *CIFAR-10* | | | | | | | |
| SVHN | 88.6±0.0 | 73.7±0.0 | 90.1±0.0 | 89.7±0.3 | 89.3±0.3 | 89.3±0.4 | **96.3±0.4** |
| LSUN-CR | 94.6±0.0 | 93.0±0.1 | 96.4±0.0 | 95.3±0.3 | 94.9±0.2 | 95.2±0.5 | **99.7±0.1** |
| CIFAR-100 | 84.1±0.0 | 79.0±0.1 | **86.2±0.0** | 84.8±0.1 | 84.5±0.0 | 84.7±0.1 | 84.2±0.3 |
| Uniform Noise | 97.5±0.0 | 66.1±0.1 | **99.6±0.0** | 97.7±0.1 | 97.7±0.1 | 95.2±0.5 | 98.4±0.4 |
| Asymptotic | 6.5±0.0 | 86.3±0.1 | **98.7±0.0** | 80.2±1.8 | 64.7±3.2 | 84.5±1.5 | 96.5±0.6 |
| *SVHN* | | | | | | | |
| CIFAR-10 | 96.4±0.0 | 97.3±0.0 | **97.6±0.0** | 97.0±0.0 | 96.8±0.0 | 96.7±0.0 | 95.8±0.5 |
| LSUN-CR | 97.1±0.0 | **98.7±0.0** | 98.6±0.0 | 96.9±0.1 | 96.9±0.1 | 96.9±0.0 | 96.4±0.3 |
| CIFAR-100 | 95.9±0.0 | 96.7±0.0 | **97.1±0.0** | 96.7±0.0 | 96.4±0.0 | 96.3±0.0 | 95.3±0.5 |
| Uniform Noise | 98.2±0.0 | **99.6±0.0** | 99.5±0.0 | 98.3±0.1 | 98.2±0.0 | 98.1±0.0 | 99.1±0.2 |
| Asymptotic | 15.6±0.0 | 79.6±0.1 | 72.5±0.0 | **99.1±0.1** | 97.8±0.1 | 97.1±0.1 | **99.1±0.3** |
| *CIFAR-100* | | | | | | | |
| SVHN | 67.5±0.0 | 73.3±0.1 | 72.9±0.0 | 70.1±1.1 | 69.0±0.4 | 69.4±0.6 | **80.1±0.7** |
| LSUN-CR | 76.6±0.0 | 80.1±0.3 | 69.2±0.0 | 70.9±1.9 | 72.4±0.6 | 72.4±1.0 | **94.6±1.3** |
| CIFAR-10 | 65.6±0.0 | **69.4±0.1** | 69.4±0.0 | 67.1±0.3 | 66.3±0.1 | 66.5±0.1 | 66.3±0.2 |
| Uniform Noise | 52.0±0.0 | **82.7±0.1** | 68.7±0.0 | 49.1±4.3 | 48.2±0.9 | 49.5±2.5 | 60.5±3.4 |
| Asymptotic | 1.6±0.0 | 66.8±0.1 | 92.3±0.0 | 93.5±1.0 | 89.9±1.1 | 91.6±0.7 | **97.3±0.3** |

## 5.1 IMAGE CLASSIFICATIONS

Here, we aim to show that LULA units and the proposed training procedure are (i) a significantly better method for tuning the uncertainty of a LA than previous methods and (ii) able to make a vanilla LA better than *non-post-hoc* (thus more expensive) UQ methods. For the purpose of (i), we compare

Table 3: Robustness to dataset shifts on the corrupted CIFAR-10 dataset (Hendrycks & Dietterich, 2019), following Ovadia et al. (2019). All values are averages and standard deviations over all perturbation types and intensities (for total of 95 dataset shifts). For accuracy, higher is better, while for ECE and NLL, lower is better.

| Metrics | MAP | MCD | DE | KFL | KFL+LL | KFL+OOD | *KFL+LULA* |
|---------|-----|-----|-----|-----|--------|---------|------------|
| Acc. | 29.4±9.0 | **30.3±9.1** | 30.0±9.8 | 29.3±9.0 | 29.4±9.0 | 29.2±8.9 | 29.1±8.9 |
| ECE | 47.3±7.2 | **22.7±9.1** | 35.6±8.1 | 36.9±7.3 | 42.0±7.2 | 30.5±7.4 | 28.9±6.6 |
| NLL | 4.067±1.09 | **2.347±0.53** | 3.321±0.97 | 3.410±0.91 | 3.721±1.00 | 3.061±0.80 | 3.028±0.72 |

LULA-augmented KFL (**KFL+LULA**) against the vanilla KFL with exact prior precision (**KFL**), KFL where the prior precision is found by maximizing validation log-likelihood (**KFL+LL**; Ritter et al., 2018b), and KFL with OOD training (**KFL+OOD**; Kristiadi et al., 2020). Moreover, for the purpose of (ii), we also compare our method with the following baselines, which have been found to yield strong results in UQ (Ovadia et al., 2019): Monte-Carlo dropout (**MCD**; Gal & Ghahramani, 2016) and deep ensemble (**DE**; Lakshminarayanan et al., 2017).

We use 5- and 8-layer CNNs for MNIST and CIFAR-10, SVHN, CIFAR-100, respectively. These networks achieve around 99%, 90%, and 50% accuracies for MNIST, CIFAR-10, and CIFAR-100, respectively. For MC-integration during the predictions, we use 10 posterior samples. We quantify the results using the standard metrics: the mean-maximum-confidence (MMC) and area-under-ROC (AUR) metrics (Hendrycks & Gimpel, 2017). All results reported are averages over ten prediction runs. Finally, we use standard test OOD datasets along with the "asymptotic" dataset introduced by Kristiadi et al. (2020) where random uniform noise images are scaled with a large number (5000).

For simplicity, we add LULA units and apply only at the last layer of the network. For each in-distribution dataset, the training of the free parameters of LULA units is performed for a single epoch using Adam over the respective validation dataset. The number of these units is obtained via a grid-search over the set $\{64, 128, 256, 512\}$, balancing both the in- and out-distribution confidences (see Appendix B for the details).

**OOD Detection** The results for MMC and AUR are shown in Table 1 and Table 2, respectively. First, we would like to direct the reader's attention to the last four columns of the table. We can see that in general KFL+LULA performs the best among all LA tuning methods. These results validate the effectiveness of the additional flexibility given by LULA units and the proposed training procedure. Indeed, without this additional flexibility, OOD training on just the prior precision becomes less effective, as shown by the results of KFL+OOD. Finally, as we can see in the results on the Asymptotic OOD dataset, LULA makes KFL significantly better at mitigating overconfidence far away from the training data. Now, compared to the contemporary baselines (MCD, DE), we can see that the vanilla KFL yields somewhat worse results. Augmenting the base KFL with LULA makes it competitive to MCD and DE in general. Keep in mind that both KFL and LULA are *post-hoc* methods.

**Dataset Shift** We use the corrupted CIFAR-10 dataset (Hendrycks & Dietterich, 2019) for measuring the robustness of LULA-augmented LA to dataset shifts, following Ovadia et al. (2019). Note that dataset shift is a slightly different concept to OOD data: it concerns about small perturbations of the true data, while OOD data are data that do not come from the true data distribution. Intuitively, humans perceive that the data under a dataset shift lies near the true data while OOD data are farther away. We present the results in Table 3. Focusing first on the last four columns in the table, we see that LULA yields the best results compared other tuning methods for KFL. Furthermore, we see that KFL+LULA outperforms DE, which has been shown by Ovadia et al. (2019) to give the state-of-the-art results in terms of robustness to dataset shifts. Finally, while MCD achieve the best results in this experiment, considering its performance in the previous OOD experiment, we draw a conclusion that KFL+LULA provides a more consistent performance over different tasks.

**Comparison with DPN** Finally we compare KFL+LULA with the (non-Bayesian) Dirichlet prior network (**DPN**, Malinin & Gales, 2018) in the Rotated-MNIST benchmark (Ovadia et al., 2019)

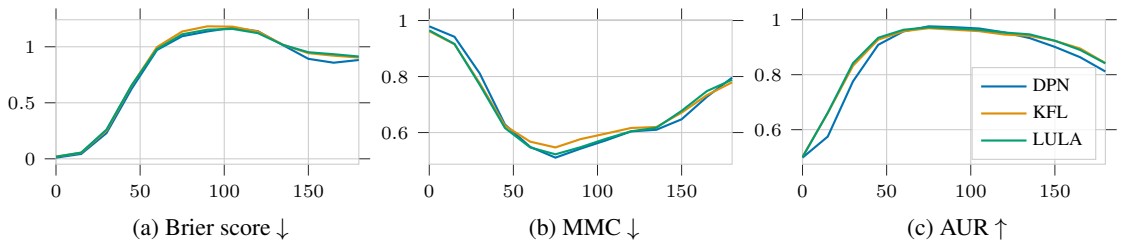

Figure 3: LULA compared to DPN on the Rotated-MNIST benchmark.

(Figure 3). We found that LULA makes the performance of KFL competitive to DPN. We stress that KFL and LULA are *post-hoc* methods, while DPN requires training from scratch.

## 5.2 COST ANALYSIS

The cost of constructing a LULA network is negligible even for our deepest network: on both the 5-and 8-layer CNNs, the wall-clock time required (with a standard consumer GPU) to add additional LULA units is on average $0.01$ seconds (over ten trials). For training, using the last-layer LA as a proxy of the true LA posterior, it took on average $7.25$ seconds and $35$ seconds for MNIST and CIFAR-10, SVHN, CIFAR-100, respectively. This tuning cost is cheap relative to the training time of the base network, which ranges between several minutes to more than an hour. We refer the reader to Table 9 (Appendix C) for the detail. All in all, LULA is not only effective, but also cost-efficient.

## 6 CONCLUSION

We have proposed LULA units: hidden units that can be added to any pre-trained MAP network for the purpose of exclusively tuning the uncertainty of a Laplace approximation without affecting its predictive performance. They can be trained via an objective that depends on both inlier and outlier datasets to minimize (resp. maximize) the network's output variance, bringing the state-of-the-art technique from non-Bayesian robust learning literature to the Bayesian world. Even with very simple outlier dataset for training, we show in extensive experiments that LULA units provide more effective *post-hoc* uncertainty tuning for Laplace approximations and make their performance competitive to more expensive baselines which require re-training of the whole network.

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

## APPENDIX A    PROOFS

**Proposition 1.** *Let $f : \mathbb{R}^n \times \mathbb{R}^d \to \mathbb{R}^k$ be a MAP-trained L-layer network parametrized by $\theta_{\mathrm{MAP}}$. Suppose $\widetilde{f} : \mathbb{R}^n \times \mathbb{R}^{\widetilde{d}} \to \mathbb{R}$ and $\widetilde{\theta}_{\mathrm{MAP}} \in \mathbb{R}^{\widetilde{d}}$ are obtained via the previous construction. For any input $x \in \mathbb{R}^n$, we have $\widetilde{f}(x; \widetilde{\theta}_{\mathrm{MAP}}) = f(x; \theta_{\mathrm{MAP}})$.*

*Proof.* Let $x \in \mathbb{R}^n$ be arbitrary. For each layer $l = 1, \dots, L$ we denote the hidden units and pre-activations of $\widetilde{f}$ as $\widetilde{h}^{(l)}$ and $\widetilde{a}^{(l)}$, respectively. We need to show that the output of $\widetilde{f}$, i.e. the last pre-activations $\widetilde{a}^{(L)}$, is equal to the last pre-activations $a^{(L)}$ of $f$.

For the first layer, we have that

$$\widetilde{a}^{(1)} = \widetilde{W}^{(1)} x + \widetilde{b}^{(1)} = \begin{pmatrix} W^{(1)} \\ \widehat{W}^{(1)} \end{pmatrix} x + \begin{pmatrix} b^{(1)} \\ \widehat{b}^{(1)} \end{pmatrix} = \begin{pmatrix} W^{(1)} x + b^{(1)} \\ \widehat{W}^{(1)} x + \widehat{b}^{(1)} \end{pmatrix} =: \begin{pmatrix} a^{(1)} \\ \widehat{a}^{(1)} \end{pmatrix} .$$

For every layer $l = 1, \dots, L-1$, we denote the hidden units as the block vector

$$\widetilde{h}^{(l)} = \begin{pmatrix} \varphi(a^{(l)}) \\ \varphi(\widehat{a}^{(l)}) \end{pmatrix} = \begin{pmatrix} h^{(l)} \\ \widehat{h}^{(l)} \end{pmatrix} .$$

Now, for the intermediate layer $l = 2, \dots, L-1$, we observe that

$$\begin{aligned} \widetilde{a}^{(l)} = \widetilde{W}^{(l)} \widetilde{h}^{(l-1)} + \widetilde{b}^{(l)} &= \begin{pmatrix} W^{(l)} & 0 \\ \widehat{W}_1^{(l)} & \widehat{W}_2^{(l)} \end{pmatrix} \begin{pmatrix} h^{(l-1)} \\ \widehat{h}^{(l-1)} \end{pmatrix} + \begin{pmatrix} b^{(l)} \\ \widehat{b}^{(l)} \end{pmatrix} \\ &= \begin{pmatrix} W^{(l)} h^{(l-1)} + 0 + b^{(l)} \\ \widehat{W}_1^{(l)} h^{(l-1)} + \widehat{W}_2^{(l)} \widehat{h}^{(l-1)} + \widehat{b}^{(l)} \end{pmatrix} =: \begin{pmatrix} a^{(l)} \\ \widehat{a}^{(l)} \end{pmatrix} . \end{aligned}$$

Finally, for the last-layer, we get

$$\widetilde{a}^{(L)} = \widetilde{W}^{(L)} x + \widetilde{b}^{(L)} = \begin{pmatrix} W^{(L)} & 0 \end{pmatrix} \begin{pmatrix} h^{(L-1)} \\ \widehat{h}^{(L-1)} \end{pmatrix} + b^{(L)} = W^{(L)} h^{(L-1)} + 0 + b^{(L)} = a^{(L)} .$$

This ends the proof. □

**Proposition 2.** *Suppose $f : \mathbb{R}^n \times \mathbb{R}^d \to \mathbb{R}$ is a real-valued network and $\widetilde{f}$ is as constructed above. Suppose further that diagonal Laplace-approximated posteriors $\mathcal{N}(\theta \mid \theta_{\mathrm{MAP}}, \mathrm{diag}(\sigma))$, $\mathcal{N}(\widetilde{\theta} \mid \widetilde{\theta}, \mathrm{diag}(\widetilde{\sigma}))$ are employed. Using the linearization (5), for any input $x \in \mathbb{R}^n$, the variance over the output $\widetilde{f}(x; \widetilde{\theta})$ is at least that of $f(x; \theta)$.*

*Proof.* W.l.o.g. we arrange the parameters $\widetilde{\theta} := (\theta^\top, \widehat{\theta}^\top)^\top$ where $\widehat{\theta} \in \mathbb{R}^{\widetilde{d}-d}$ contains the weights corresponding to the the additional LULA units. If $g(x)$ is the gradient of the output $f(x; \theta)$ w.r.t. $\theta$, then the gradient of $\widetilde{f}(x; \widetilde{\theta})$ w.r.t. $\widetilde{\theta}$, say $\widetilde{g}(x)$, can be written as the concatenation $(g(x)^\top, \widehat{g}(x)^\top)^\top$ where $\widehat{g}(x)$ is the corresponding gradient of $\widehat{\theta}$. Furthermore, $\mathrm{diag}(\widetilde{\sigma})$ has diagonal elements

$$\left( \sigma_{11}, \dots, \sigma_{dd}, \widehat{\sigma}_{11}, \dots, \widehat{\sigma}_{\widetilde{d}-d, \widetilde{d}-d} \right)^\top =: (\sigma^\top, \widehat{\sigma}^\top)^\top .$$

Hence we have

$$\begin{aligned} \widetilde{v}(x) &= \widetilde{g}(x)^\top \mathrm{diag}(\widetilde{\sigma}) \widetilde{g}(x) \\ &= \underbrace{g(x)^\top \mathrm{diag}(\sigma) g(x)}_{=v(x)} + \widehat{g}(x)^\top \mathrm{diag}(\widehat{\sigma}) \widehat{g}(x) \\ &\geq v(x) , \end{aligned}$$

since $\mathrm{diag}(\widehat{\sigma})$ is positive-definite. □

---

**Algorithm 2** Adding LULA units.

**Input:**

$L$-layer net with a MAP estimate $\theta_{\text{MAP}} = (W_{\text{MAP}}^{(l)}, b_{\text{MAP}}^{(l)})_{l=1}^L$. Sequence of non-negative integers $(m_l)_{l=1}^L$.

1: **for** $l = 1, \ldots, L-1$ **do**

2: $\quad$ vec $\widehat{W}^{(l)} \sim p(\text{vec } \widehat{W}^{(l)})$ $\qquad\qquad\qquad\qquad\qquad\qquad$ ▷ Draw from a prior

3: $\quad$ $\widehat{b}^{(l)} \sim p(\widehat{b}^{(l)})$ $\qquad\qquad\qquad\qquad\qquad\qquad\qquad\qquad$ ▷ Draw from a prior

4: $\quad$ $\widetilde{W}_{\text{MAP}}^{(l)} = \begin{pmatrix} W_{\text{MAP}}^{(l)} & 0 \\ \widehat{W}_1^{(l)} & \widehat{W}_2^{(l)} \end{pmatrix}$ $\qquad\qquad$ ▷ The zero submatrix 0 is of size $n_l \times m_{l-1}$

5: $\quad$ $\widetilde{b}_{\text{MAP}}^{(l)} := \begin{pmatrix} b_{\text{MAP}}^{(l)} \\ \widehat{b}^{(l)} \end{pmatrix}$

6: **end for**

7: $\widetilde{W}_{\text{MAP}}^{(L)} = (W_{\text{MAP}}^{(L)}, 0)$ $\qquad\qquad\qquad\qquad$ ▷ The zero submatrix is of size $k \times m_{L-1}$

8: $\widetilde{b}_{\text{MAP}}^{(L)} = b_{\text{MAP}}^{(L)}$

9: $\widetilde{\theta}_{\text{MAP}} = (\widetilde{W}_{\text{MAP}}^{(l)}, \widetilde{b}_{\text{MAP}}^{(l)})_{l=1}^L$

10: **return** $\widetilde{\theta}_{\text{MAP}}$

---

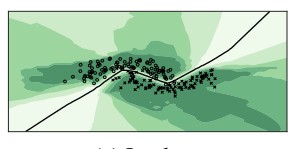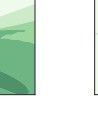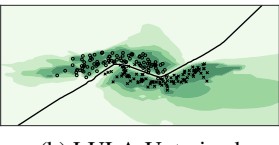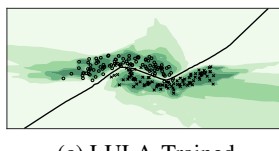

$\qquad$ (a) Laplace $\qquad\qquad\qquad$ (b) LULA-Untrained $\qquad\qquad\qquad$ (c) LULA-Trained

Figure 4: Even when their weights are assigned randomly, LULA units improve the vanilla Laplace in terms of UQ. Fine-tuning the LULA weights improve it even further, in particular in terms of confidence far from the data—trained, LULA yields less confident prediction in this region.

## APPENDIX B    IMPLEMENTATION

We summarize the augmentation of a network with LULA units in Algorithm 2. Note that the priors of the free parameters $\widehat{W}^{(l)}$, $\widehat{b}^{(l)}$ (lines 2 and 3) can be chosen as independent Gaussians—this reflects the standard procedure for initializing NNs' parameters.

We train LULA units for a single epoch (since for each dataset, we have a large amount of training points) with learning rate $0.01$. For each dataset, the number of the additional last-layer units $m_L$ is obtained via a grid search over the set $\{64, 128, 256, 512\} =: M_L$, minimizing the absolute distance to the optimal MMC, i.e. $1$ and $1/k$ for the in- and out-distribution validation set, respectively:

$$m_L = \arg\min_{m \in M_L} |1 - \text{MMC}_{\text{in}}(m)| + |1/k - \text{MMC}_{\text{out}}(m)|, \qquad (9)$$

where $\text{MMC}_{\text{in}}$ and $\text{MMC}_{\text{out}}$ are the validation in- and out-distribution MMC of the Laplace-approximated, trained-LULA, respectively.

## APPENDIX C    ADDITIONAL EXPERIMENT RESULTS

### C.1    TOY DATASET

In practice, one can simply set $\{\widehat{W}^{(l)}, \widehat{b}^{(l)}\}_{l=1}^{L-1}$ randomly given a prior, e.g. the standard Gaussian (see also Algorithm 2). To validate this practice, we show the vanilla Laplace, untrained LULA,

Table 4: Predictive performances on UCI regression datasets in term of average test log-likelihood. The numbers reported are averages over ten runs along the corresponding standard deviations. The performances of LULA are similar to KFL's. The differences between their exact values are likely due to MC-integration.

| Dataset | MAP | MCD | DE | KFL | KFL+LULA |
|---|---|---|---|---|---|
| Housing | -2.794±0.012 | -2.731±0.031 | -3.045±0.009 | -3.506±0.055 | -3.495±0.047 |
| Concrete | -3.409±0.036 | -4.370±0.001 | -3.951±0.062 | -4.730±0.205 | -4.365±0.094 |
| Energy | -2.270±0.128 | -2.120±0.104 | -2.673±0.015 | -2.707±0.030 | -2.698±0.014 |
| Kin8nm | -0.923±0.000 | -0.926±0.000 | 1.086±0.022 | -0.965±0.003 | -0.969±0.003 |
| Power | -3.154±0.002 | -4.185±0.003 | -54.804±7.728 | -3.273±0.015 | -3.277±0.024 |
| Wine | -1.190±0.014 | -1.185±0.005 | -1.038±0.018 | -1.624±0.075 | -1.630±0.092 |
| Yacht | -1.835±0.053 | -1.724±0.117 | -3.272±0.079 | -2.509±0.367 | -2.663±0.276 |

Table 5: UQ performances on UCI datasets. Values are the average (over all data points and ten trials) predictive standard deviations. Lower is better for test data and vice-versa for outliers. By definition, MAP does not have (epistemic) uncertainty.

| | Test set ↓ | | | | Outliers ↑ | | | |
|---|---|---|---|---|---|---|---|---|
| Dataset | MCD | DE | KFL | *KFL+LULA* | MCD | DE | KFL | *KFL+LULA* |
| Housing | 1.40 | 5.82 | **1.26** | 1.37 | 61.49 | 145.33 | 222.76 | **377.92** |
| Concrete | **1.00** | 8.11 | 10.44 | 16.89 | 4.06 | 964.63 | 30898.92 | **83241.42** |
| Energy | 1.32 | 4.40 | 1.05 | 1.08 | 45.69 | 126.11 | 1070.09 | **5163.53** |
| Kin8nm | **0.04** | 0.10 | 0.14 | 0.18 | 0.24 | **2.12** | 0.80 | **2.12** |
| Power | 14.73 | 19.85 | **2.85** | 3.20 | 273.08 | 12235.87 | 4148.98 | **221287.80** |
| Wine | **0.24** | 0.64 | 1.15 | 1.22 | 0.30 | 28.57 | 186.76 | **21383.17** |
| Yacht | **0.54** | 5.17 | 2.08 | 2.78 | 81.98 | 187.41 | 5105.69 | **13119.99** |

and fine-tuned LULA's confidence in Figure 4. Even when set randomly from the standard Gaussian prior, LULA weights provide a significant improvement over the vanilla Laplace. Moreover, training them yields even better predictive confidence estimates. In particular, far from the data, the confidence becomes even lower, while still maintaining high confidence in the data regions.

## C.2 UCI REGRESSION

To validate the performance of LULA in regressions, we employ a subset of the UCI regression benchmark datasets. Following previous works, the network architecture used here is a single-hidden-layer ReLU network with 50 hidden units. The data are standardized to have zero mean and unit variance. We use 50 LULA units and optimize them for 40 epochs using OOD data sampled uniformly from $[-10, 10]^n$. For MCD, KFL, and KFL+LULA, each prediction is done via MC-integration with 100 samples. For the evaluation of each dataset, we use a 60-20-20 train-validation-test split. We repeat each train-test process 10 times and take the average.

In Table 5 we report the average predictive standard deviation for each dataset. Note that this metric is the direct generalization of the 1D error bar in Figure 1 (top) to multi dimension. The outliers are sampled uniformly from $[-10, 10]^n$. Note that since the inlier data are centered around the origin and have unit variance, they lie approximately in a Euclidean ball with radius 2. Therefore, these outliers are very far away from them. Thus, naturally, high uncertainty values over these outliers are desirable. Uncertainties over the test sets are generally low for all methods, although KFL+LULA has slightly higher uncertainties compared to the base KFL. However, KFL+LULA yield much higher uncertainties over outliers across all datasets, significantly more than the baselines. Moreover, in Table 4, we show that KFL+LULA maintains the predictive performance of the base KFL. Altogether, they imply that KFL+LULA can detect outliers better than other methods without costing the predictive performance.

## C.3 IMAGE CLASSIFICATION

In Table 6 we present the sensitivity analysis of confidences under a Laplace approximation w.r.t. the number of additional LULA units. Generally, we found that small number of additional LULA units,

Table 6: In- and out-distribution validation MMCs for varying numbers of additional LULA units. "In" and "out" values are in percent. "Loss" is the value of the loss in (9). Missing entries signify that errors occurred, see Section C.3 for details.

| # units | MNIST In ↑ | Out ↓ | Loss ↓ | CIFAR-10 In ↑ | Out ↓ | Loss ↓ | CIFAR-100 In ↑ | Out ↓ | Loss ↓ |
|---|---|---|---|---|---|---|---|---|---|
| 16 | 97.1 | 46.3 | 0.392 | 91.3 | 21.6 | 0.203 | 54.0 | 12.8 | 0.578 |
| 32 | 97.1 | 19.7 | 0.127 | 90.4 | 17.7 | 0.173 | 50.5 | 8.7 | 0.572 |
| 64 | 97.0 | 27.0 | 0.200 | 89.3 | 19.6 | 0.203 | 43.8 | 7.8 | 0.629 |
| 128 | 96.7 | 28.9 | 0.223 | 83.9 | 18.2 | 0.243 | 31.0 | 6.5 | 0.745 |
| 256 | 94.2 | 18.2 | 0.140 | 62.1 | 18.4 | 0.463 | 14.3 | 6.2 | 0.909 |
| 512 | 68.4 | 17.6 | 0.392 | - | - | - | - | - | - |

Table 7: Accuracies (in percent) over image classification test sets. Values are averages over ten trials.

| Dataset | MAP | MCD | DE | KFL | KFL+LULA |
|---|---|---|---|---|---|
| MNIST | 99.1±0.0 | 99.1±0.0 | 99.1±0.0 | 99.1±0.0 | 99.1±0.0 |
| CIFAR-10 | 90.1±0.0 | 84.4±0.0 | 91.2±0.0 | 89.9±0.1 | 89.5±0.1 |
| CIFAR-100 | 53.2±0.0 | 60.5±0.1 | 59.6±0.0 | 52.2±0.2 | 51.3±0.4 |

e.g. 32 and 64, is optimal. It is clear that increasing the number of LULA units decreases both the in- and out-distribution confidences. In the case of larger networks, we found that larger values (e.g. 512 in CIFAR-10) make the Hessian badly conditioned, resulting in numerical instability during its inversion. One might be able to alleviate this issue by additionally tuning the prior precision hyperparameter of the Laplace approximation (as in (Ritter et al., 2018b; Kristiadi et al., 2020)), which corresponds to varying the strength of the diagonal correction of the Hessian. However, we emphasize that even with small amounts of additional LULA units, we can already improve vanilla Laplace approximations significantly, as shown in the main text (Section 5.1).

We present the predictive performances of all methods in Table 7. LULA achieves similar accuracies to the base MAP and KFL baselines. Differences in their exact values likely due to various approximations used (e.g. MC-integral). In the case of CIFAR-100, we found that MAP underperforms compared to MCD and DE. This might be because of overfitting, since only weight decay is used for regularization, in contrast to MCD where dropout is used on top of weight decay. Due to MAP's underperformance, LULA also underperform. However, we stress that whenever the base MAP model performs well, by construction LULA will also perform well.

As a supplement we show the performance of KFL+LULA against DPN in OOD detection on MNIST (Table 8). We found that KFL+LULA performance on OOD detection are competitive or better than DPN.

## C.4 COMPUTATIONAL COST

To supplement the cost analysis in the main text, we show the wall-clock times required for the construction and training of LULA units in Table 9.

Table 8: LULA compared to DPN on OOD detection in terms of MMC and AUR, both in percent.

| Train - Test | MMC DPN | KFL+LULA | AUR ↑ DPN | KFL+LULA |
|---|---|---|---|---|
| *MNIST* | | | | |
| E-MNIST | 74.5±0.0 | 71.4±0.4 | 87.7±0.0 | **88.3±0.2** |
| F-MNIST | 58.5±0.0 | 49.4±1.9 | 96.8±0.0 | **98.0±0.4** |
| Smooth Noise | 10.5±0.0 | 22.2±0.6 | **100.0±0.0** | **100.0±0.0** |
| Uniform Noise | 61.0±0.0 | 33.8±2.5 | 98.2±0.0 | **99.8±0.1** |
| Asymptotic | 100.0±0.0 | 46.0±2.6 | 0.2±0.0 | **99.0±0.2** |

Table 9: Wall-clock time of adding and training LULA units. All values are in seconds.

| Dataset | Construction | Training |
|---------|--------------|---------|
| MNIST | 0.010±0.000 | 7.252±0.033 |
| CIFAR-10 | 0.011±0.003 | 35.108±0.243 |
| CIFAR-100 | 0.009±0.000 | 34.913±0.253 |

Table 10: Average confidences (MMCs in percent) on 20-layer CNNs over ten prediction runs. Lower is better for OOD data. Values shown for each in-distribution dataset are ECE—lower is better. Underline and bold faces indicate the highest values over the last four columns and all columns in a given row, respectively.

| Train - Test | MAP | MCD | DE | KFL | KFL+LL | KFL+OOD | *KFL+LULA* |
|--------------|-----|-----|-----|-----|--------|---------|------------|
| *CIFAR-10* | 15.0±0.0 | 6.8±0.3 | **4.2±0.0** | 9.9±0.4 | 5.8±0.8 | 6.1±0.6 | 5.2±0.5 |
| SVHN | 70.7±0.0 | 67.3±0.0 | 58.3±0.0 | 65.2±0.6 | 60.1±1.2 | 60.5±1.5 | **57.5±0.9** |
| LSUN-CR | 56.2±0.0 | 56.0±0.2 | 51.3±0.0 | 51.3±0.8 | 45.5±1.4 | 47.3±1.8 | **39.8±1.4** |
| CIFAR-100 | 78.3±0.0 | 69.9±0.0 | 66.5±0.0 | 72.9±0.3 | 67.6±0.8 | 68.3±0.8 | **66.2±0.6** |
| Uniform Noise | 84.0±0.0 | 74.8±0.0 | **54.8±0.0** | 74.2±1.5 | 63.3±2.6 | 65.2±3.2 | 59.5±2.3 |
| Asymptotic | 99.9±0.0 | 95.2±0.0 | **70.6±0.0** | 87.1±1.4 | 80.4±2.4 | 82.1±2.7 | 73.0±1.6 |
| *SVHN* | 10.5±0.0 | **2.7±0.2** | 17.3±0.0 | 6.3±0.2 | 5.8±0.4 | 4.1±0.4 | 3.4±0.3 |
| CIFAR-10 | 73.8±0.0 | 55.2±0.0 | **43.9±0.0** | 67.9±0.2 | 67.4±0.4 | 64.7±0.5 | 59.3±0.9 |
| LSUN-CR | 64.8±0.0 | 50.6±0.1 | **42.0±0.0** | 61.8±0.3 | 61.5±0.4 | 59.7±0.3 | 45.5±1.5 |
| CIFAR-100 | 74.8±0.0 | 56.1±0.0 | **45.0±0.0** | 69.4±0.3 | 68.9±0.4 | 66.2±0.4 | 60.7±0.9 |
| Uniform Noise | 56.6±0.0 | 47.2±0.0 | **36.5±0.0** | 54.1±0.2 | 53.8±0.3 | 52.4±0.3 | 41.0±1.7 |
| Asymptotic | 99.8±0.0 | 65.5±0.0 | **44.4±0.0** | 84.1±0.6 | 83.5±0.7 | 79.2±0.5 | 69.1±1.6 |
| *CIFAR-100* | 38.7±0.0 | 20.6±0.1 | 10.8±0.0 | 21.5±0.3 | 14.6±0.3 | 10.4±0.2 | **4.6±0.2** |
| SVHN | 69.3±0.0 | 53.5±0.0 | 47.1±0.0 | 53.5±0.4 | 46.8±0.4 | 42.8±0.4 | **26.7±0.5** |
| LSUN-CR | 67.1±0.0 | 52.6±0.3 | 46.0±0.0 | 52.8±0.5 | 46.3±0.6 | 42.6±0.7 | **23.4±1.1** |
| CIFAR-10 | 73.5±0.0 | 57.9±0.1 | 52.2±0.0 | 59.3±0.2 | 53.1±0.1 | 49.0±0.3 | **40.7±0.3** |
| Uniform Noise | 79.9±0.0 | 52.4±0.1 | 47.3±0.0 | 61.2±1.1 | 52.2±1.3 | 48.7±1.6 | **25.3±1.6** |
| Asymptotic | 97.4±0.0 | 48.1±0.1 | 41.7±0.0 | 52.8±1.3 | 45.0±1.7 | 39.4±1.2 | **37.8±1.2** |

## C.5 DEEPER NETWORKS

We also asses the performance of LULA in larger network. We use a 20-layer CNN on CIFAR-10, SVHN, and CIFAR-100. Both the KFL and LULA are applied only at the last-layer of the network. The results, in terms of MMC, expected calibration error (ECE), and AUR, are presented in Table 10 and Table 11. We observe that LULA is the best method for uncertainty tuning in LA: It makes KFL better calibrated in both in- and out-distribution settings. Moreover, the LULA-imbued KFL is competitive to DE, which has been shown by Ovadia et al. (2019) to be the best Bayesian method for uncertainty quantification. Note that, KFL+LULA is a *post-hoc* method and thus can be applied to any pre-trained network. In contrast, DE requires training multiple networks (usually 5) from scratch which can be very expensive.

We additionally show the performance of LULA when applied on top of a KFL-approximated DenseNet-121 (Huang et al., 2017) in Tables 12 and 13. LULA generally outperforms previous uncertainty tuning methods for LA and is competitive to DE. However, we observe in SVHN that LULA do not improve KFL significantly. This issue is due to the usage the Smooth Noise dataset, which has already assigned low confidence in this case, for training LULA. Thus, we re-train LULA with the Uniform Noise dataset and present the result in Table 14. We show that using this dataset, we obtain better OOD calibration performance, outperforming DE.

Table 11: OOD detection performance measured by the AUR metric on 20-layer CNNs. Values reported are averages over ten prediction runs. Higher is better. Underline and bold faces indicate the highest values over the last four columns and all columns in a given row, respectively.

| Train - Test | MAP | MCD | DE | KFL | KFL+LL | KFL+OOD | KFL+LULA |
|---|---|---|---|---|---|---|---|
| *CIFAR-10* | | | | | | | |
| SVHN | 91.4±0.0 | 89.0±0.0 | **92.8±0.0** | 90.9±0.2 | 90.6±0.3 | 90.8±0.3 | 90.9±0.2 |
| LSUN-CR | 95.6±0.0 | 93.4±0.1 | 94.8±0.0 | 95.5±0.2 | 96.0±0.3 | 95.6±0.4 | **97.0±0.3** |
| CIFAR-100 | 85.1±0.0 | 85.5±0.0 | **87.2±0.0** | 85.1±0.1 | 85.1±0.1 | 85.1±0.1 | 85.2±0.1 |
| Uniform Noise | 86.0±0.0 | 85.9±0.0 | **94.7±0.0** | 87.4±0.7 | 89.7±0.9 | 89.2±1.2 | 90.3±0.8 |
| Asymptotic | 12.7±0.0 | 35.0±0.3 | 77.9±0.0 | 56.9±4.4 | 70.5±5.9 | 67.2±6.9 | **81.0±1.7** |
| *SVHN* | | | | | | | |
| CIFAR-10 | 93.2±0.0 | 96.4±0.0 | **97.2±0.0** | 93.9±0.0 | 94.0±0.0 | 94.2±0.1 | 93.9±0.2 |
| LSUN-CR | 96.4±0.0 | 97.4±0.0 | **98.1±0.0** | 96.1±0.0 | 96.1±0.1 | 96.0±0.1 | 96.7±0.1 |
| CIFAR-100 | 92.9±0.0 | 96.1±0.0 | **96.8±0.0** | 93.5±0.0 | 93.5±0.0 | 93.7±0.1 | 93.5±0.2 |
| Uniform Noise | 98.1±0.0 | 98.0±0.0 | **99.0±0.0** | 97.8±0.0 | 97.8±0.0 | 97.7±0.0 | 98.1±0.1 |
| Asymptotic | 6.9±0.0 | 94.7±0.0 | **95.4±0.0** | 59.0±1.5 | 60.6±1.8 | 69.3±1.1 | 83.2±2.0 |
| *CIFAR-100* | | | | | | | |
| SVHN | 69.8±0.0 | 70.5±0.1 | 73.3±0.0 | 71.2±0.5 | 72.2±0.4 | 72.5±0.4 | **83.7±0.5** |
| LSUN-CR | 72.1±0.0 | 71.3±0.3 | 74.4±0.0 | 71.9±0.6 | 72.7±0.6 | 72.7±0.8 | **88.3±1.3** |
| CIFAR-10 | 64.7±0.0 | 65.7±0.1 | **67.9±0.0** | 64.8±0.2 | 65.0±0.1 | 65.1±0.2 | 64.3±0.3 |
| Uniform Noise | 58.2±0.0 | 72.0±0.1 | 73.1±0.0 | 63.2±1.0 | 66.1±1.5 | 65.2±1.9 | **85.9±2.2** |
| Asymptotic | 13.6±0.0 | 76.1±0.1 | **78.6±0.0** | 72.1±1.4 | 74.2±2.0 | 76.6±1.5 | 67.6±1.8 |

Table 12: Average confidences (MMCs in percent) on DenseNet-121 over ten prediction runs. Lower is better for OOD data. Values shown for each in-distribution dataset are ECE—lower is better. Underline and bold faces indicate the highest values over the last four columns and all columns in a given row, respectively.

| Train - Test | MAP | MCD | DE | KFL | KFL+LL | KFL+OOD | KFL+LULA |
|---|---|---|---|---|---|---|---|
| *CIFAR-10* | 23.3±0.0 | 10.2±0.5 | **8.1±0.0** | 21.2±0.4 | 16.3±0.5 | 15.5±0.4 | 10.3±0.7 |
| SVHN | 87.8±0.0 | 74.3±0.0 | 67.6±0.0 | 85.1±0.2 | 80.7±0.6 | 79.8±1.1 | **30.9±1.5** |
| LSUN-CR | 80.0±0.0 | 70.2±0.2 | 63.4±0.0 | 77.0±0.4 | 72.4±1.2 | 71.8±0.7 | **34.0±2.3** |
| CIFAR-100 | 86.7±0.0 | 79.6±0.0 | **74.2±0.0** | 84.8±0.1 | 81.6±0.3 | 81.0±0.2 | 77.7±0.5 |
| Smooth Noise | 76.0±0.1 | 80.3±0.0 | 54.2±0.1 | 70.6±0.4 | 63.4±1.1 | 62.5±1.0 | **24.3±1.3** |
| Uniform Noise | 84.1±0.0 | 82.3±0.0 | 69.4±0.0 | 77.8±0.6 | 70.4±1.5 | **68.7±1.5** | 75.8±0.8 |
| Asymptotic | 100.0±0.0 | 100.0±0.0 | 20.0±0.0 | 63.8±1.5 | 44.5±1.4 | 43.4±1.9 | 58.4±1.8 |
| *SVHN* | 4.7±0.0 | **2.7±0.2** | 3.1±0.0 | 4.3±0.2 | 3.9±0.2 | 3.6±0.2 | 3.0±0.2 |
| CIFAR-10 | 69.9±0.0 | 56.1±0.0 | **50.5±0.0** | 69.3±0.1 | 68.6±0.3 | 67.8±0.3 | 67.8±0.2 |
| LSUN-CR | 64.1±0.0 | 52.3±0.2 | **51.3±0.0** | 63.8±0.1 | 63.4±0.2 | 63.0±0.2 | 61.4±0.4 |
| CIFAR-100 | 69.4±0.0 | 57.3±0.0 | **51.1±0.0** | 68.9±0.1 | 68.2±0.3 | 67.5±0.2 | 67.5±0.2 |
| Smooth Noise | 30.9±0.1 | **25.4±0.0** | 33.6±0.0 | 30.8±0.1 | 30.7±0.1 | 30.6±0.0 | 27.8±0.5 |
| Uniform Noise | 63.6±0.0 | 55.8±0.0 | **54.2±0.0** | 63.4±0.1 | 63.1±0.2 | 62.8±0.1 | 62.8±0.2 |
| Asymptotic | 100.0±0.0 | 98.8±0.0 | 40.0±0.0 | 34.1±2.1 | 26.6±1.1 | 25.7±1.2 | **22.7±0.6** |
| *CIFAR-100* | 27.3±0.0 | 19.1±0.2 | 6.2±0.0 | 5.3±0.3 | 10.0±0.2 | **4.7±0.3** | 13.3±0.4 |
| SVHN | 76.2±0.0 | 72.6±0.0 | 53.3±0.0 | 48.1±1.5 | 54.9±1.3 | 40.0±1.4 | **15.6±0.8** |
| LSUN-CR | 71.1±0.0 | 67.2±0.2 | 50.5±0.0 | 46.9±1.2 | 53.2±0.9 | 39.8±1.4 | **17.4±1.1** |
| CIFAR-10 | 77.8±0.0 | 70.2±0.0 | 59.3±0.0 | 57.7±0.2 | 63.1±0.2 | 51.2±0.1 | **40.3±0.2** |
| Smooth Noise | 93.1±0.1 | 90.4±0.1 | 76.9±0.1 | 61.9±2.2 | 70.7±1.5 | 50.3±2.6 | **14.1±1.4** |
| Uniform Noise | 79.8±0.0 | 56.1±0.1 | 53.6±0.0 | 36.5±2.5 | 43.6±3.3 | 28.5±2.7 | **22.4±1.3** |
| Asymptotic | 100.0±0.0 | 96.5±0.0 | **20.0±0.0** | 48.6±3.3 | 61.5±2.4 | 35.8±3.5 | 21.1±1.5 |

Table 13: OOD detection performance measured by the AUR metric on DenseNet-121. Values reported are averages over ten prediction runs. Higher is better. Underline and bold faces indicate the highest values over the last four columns and all columns in a given row, respectively.

| Train - Test | MAP | MCD | DE | KFL | KFL+LL | KFL+OOD | *KFL+LULA* |
|---|---|---|---|---|---|---|---|
| *CIFAR-10* | | | | | | | |
| SVHN | 88.2±0.0 | 92.5±0.0 | 94.1±0.0 | 89.5±0.2 | 90.9±0.3 | 91.2±0.7 | **99.7±0.0** |
| LSUN-CR | 94.0±0.0 | 94.2±0.1 | 95.3±0.0 | 94.3±0.1 | 94.5±0.3 | 94.5±0.2 | **99.5±0.1** |
| CIFAR-100 | 88.9±0.0 | 88.5±0.0 | **90.5±0.0** | 89.0±0.0 | 89.1±0.0 | 89.2±0.0 | 87.9±0.1 |
| Smooth Noise | 96.2±0.0 | 91.5±0.0 | 97.6±0.0 | 96.7±0.1 | 97.2±0.2 | 97.2±0.1 | **100.0±0.0** |
| Uniform Noise | 93.6±0.0 | 91.6±0.0 | 94.5±0.0 | 94.8±0.2 | 95.7±0.3 | **95.9±0.4** | 92.0±0.3 |
| Asymptotic | 26.1±0.0 | 18.0±0.1 | **100.0±0.0** | 97.9±0.2 | 99.5±0.1 | 99.5±0.1 | 97.2±0.3 |
| *SVHN* | | | | | | | |
| CIFAR-10 | 93.6±0.0 | 96.7±0.0 | **97.8±0.0** | 93.7±0.0 | 93.8±0.1 | 93.9±0.1 | 93.2±0.2 |
| LSUN-CR | 96.1±0.0 | 97.6±0.0 | **97.9±0.0** | 96.1±0.0 | 96.1±0.0 | 96.1±0.0 | 95.8±0.1 |
| CIFAR-100 | 93.3±0.0 | 96.1±0.0 | **97.4±0.0** | 93.4±0.0 | 93.4±0.1 | 93.6±0.1 | 92.9±0.2 |
| Smooth Noise | 99.4±0.0 | **99.7±0.0** | 99.6±0.0 | 99.4±0.0 | 99.4±0.0 | 99.4±0.0 | 99.6±0.0 |
| Uniform Noise | 97.1±0.0 | 97.5±0.0 | **98.0±0.0** | 97.1±0.0 | 97.0±0.0 | 97.0±0.0 | 96.6±0.1 |
| Asymptotic | 0.0±0.0 | 14.4±0.3 | 99.7±0.0 | 99.8±0.1 | 99.9±0.0 | **100.0±0.0** | **100.0±0.0** |
| *CIFAR-100* | | | | | | | |
| SVHN | 77.2±0.0 | 73.2±0.0 | 83.5±0.0 | 83.8±1.1 | 82.4±1.0 | 85.6±1.1 | **97.5±0.5** |
| LSUN-CR | 81.4±0.0 | 79.6±0.1 | 85.7±0.0 | 84.6±0.8 | 83.7±0.6 | 85.7±1.1 | **96.2±0.8** |
| CIFAR-10 | 74.8±0.0 | 75.8±0.0 | **78.6±0.0** | 75.3±0.1 | 75.2±0.2 | 75.5±0.2 | 73.9±0.3 |
| Smooth Noise | 59.4±0.1 | 54.6±0.1 | 62.0±0.1 | 71.4±1.9 | 68.1±1.2 | 76.2±2.4 | **97.7±1.2** |
| Uniform Noise | 78.7±0.0 | 88.4±0.1 | 85.1±0.0 | 93.1±1.5 | 91.5±2.0 | **94.8±1.7** | 93.1±1.2 |
| Asymptotic | 4.5±0.0 | 32.9±0.2 | **99.9±0.0** | 84.9±2.6 | 79.5±1.8 | 90.0±2.6 | 93.9±1.1 |

Table 14: LULA's OOD detection performance on DenseNet-121 with uniform noises as the training OOD data. Values are the ECE, MMC, and AUR metrics, averaged over ten prediction runs. Underline and bold faces indicate the highest values over the last four columns and all columns in a given row, respectively.

| Train - Test | MAP | MCD | DE | KFL | KFL+LL | KFL+OOD | *KFL+LULA* |
|---|---|---|---|---|---|---|---|
| *SVHN* (ECE, MMC) | 4.7±0.0 | **2.7±0.2** | 3.1±0.0 | 4.3±0.2 | 3.9±0.2 | 3.6±0.2 | 3.6 |
| CIFAR-10 | 69.9±0.0 | 56.1±0.0 | 50.5±0.0 | 69.3±0.1 | 68.6±0.3 | 67.8±0.3 | **39.1±1.8** |
| LSUN-CR | 64.1±0.0 | 52.3±0.2 | 51.3±0.0 | 63.8±0.1 | 63.4±0.2 | 63.0±0.2 | **29.6±2.3** |
| CIFAR-100 | 69.4±0.0 | 57.3±0.0 | 51.1±0.0 | 68.9±0.1 | 68.2±0.3 | 67.5±0.2 | **42.5±1.7** |
| Smooth Noise | 30.9±0.1 | **25.4±0.0** | 33.6±0.0 | 30.8±0.1 | 30.7±0.1 | 30.6±0.0 | 27.5±0.3 |
| Uniform Noise | 63.6±0.0 | 55.8±0.0 | 54.2±0.0 | 63.4±0.1 | 63.1±0.2 | 62.8±0.1 | **22.9±0.6** |
| Asymptotic | 100.0±0.0 | 98.8±0.0 | 40.0±0.0 | 34.1±2.1 | 26.6±1.1 | 25.7±1.2 | **23.3±0.6** |
| *SVHN* (AUR) | | | | | | | |
| CIFAR-10 | 93.6±0.0 | 96.7±0.0 | 97.8±0.0 | 93.7±0.0 | 93.8±0.1 | 93.9±0.1 | **98.7±0.1** |
| LSUN-CR | 96.1±0.0 | 97.6±0.0 | 97.9±0.0 | 96.1±0.0 | 96.1±0.0 | 96.1±0.0 | **99.5±0.1** |
| CIFAR-100 | 93.3±0.0 | 96.1±0.0 | 97.4±0.0 | 93.4±0.0 | 93.4±0.1 | 93.6±0.1 | **98.0±0.1** |
| Smooth Noise | 99.4±0.0 | **99.7±0.0** | 99.6±0.0 | 99.4±0.0 | 99.4±0.0 | 99.4±0.0 | 99.6±0.0 |
| Uniform Noise | 97.1±0.0 | 97.5±0.0 | 98.0±0.0 | 97.1±0.0 | 97.0±0.0 | 97.0±0.0 | **99.9±0.0** |
| Asymptotic | 0.0±0.0 | 14.4±0.3 | 99.7±0.0 | 99.8±0.1 | 99.9±0.0 | **100.0±0.0** | 99.9±0.0 |

