# OpenReview forum: "Learnable Uncertainty under Laplace Approximations"
_ICLR.cc/2021/Conference — Reject_

### Official Review · AnonReviewer2 · 2020-10-25
**Not a correct way of doing Bayesian inference**

**Rating:** 4
**Confidence:** 2

**Review:**

The paper proposes a post-hoc uncertainty tuning pipeline for Bayesian neural networks. After getting the point estimate, it adds extra dimensions to the weight matrices and hidden layers, which has no effect on the network output, with the hope that it would influence the variance of the original network weights under the Laplacian approximation. More specifically, it tunes the extra weights by optimizing another objective borrowed from the non-Bayesian robust learning literature, which encourages low uncertainty over real (extra, validation) data, and high uncertainty over manually constructed, out-of-distribution data.

Using Eq (7) and (8) to quantify posterior uncertainty doesn't seem correct to me. Instead of manually building some fake OOD data and forcing their posterior variance to be large, plus forcing all real data points' posterior variance to be small, one should focus more on the different posterior uncertainty *within* the real observed data. For example, data in highly uncertain areas should naturally have higher posterior variance. But in the proposed method, they are all forced to have small variances as long as they are real data.

Another confusion I have is more related to the math behind. The extra network weights and hidden dimensions are designed not to have any connection to generate the output. If so, does it mean that no matter what values the extra network weights take, their curvature to the output prediction should be entirely flat? How would these values affect the uncertainty if their Hessian values are zero? It would be beneficial if the authors could explain more about why that's not the case.


--------------------------
POST DISCUSSION UPDATE

The central part of this work about how the extra network weights only affect the curvature still confuses me. But I'm more motivated by the proposed objective function that borrows idea from non-Bayesian robust learning literature.

---

> ### Author Response · Authors · 2020-11-17
> **Response to AnonReviewer2**
>
> We appreciate the feedback. We hope the following could clear your concerns up.
>
>
> * **Using Eq (7) and (8) to quantify posterior uncertainty doesn't seem correct?  One should focus more on the different posterior uncertainty within the real observed data?** The idea behind the objective in eqs. 7 and 8 is a widely accepted standard procedure in non-Bayesian robust learning literature [1, 2, 3, etc.].  We believe that explicitly forcing this behavior using some OOD data is required since even if standard Bayesian NNs have been shown to be uncertain far from the data, their confidence estimates in these regions can still be rather high, or even be arbitrarily bad [4]. The OOD data used in LULA’s training helps to calibrate uncertainty/confidence in these regions.
>
> * **Does it mean that no matter what values the extra network weights take, their curvature to the output prediction should be entirely flat? How can it affect uncertainty?** We believe that you have misunderstood this aspect, which is central to the paper. LULA works in the parameter space of a network, not in the output space. Intuitively, by adding LULA units, we obtain an augmented version of the network’s loss landscape. The goal of LULA training is then to exploit the weight-space symmetry (i.e. different parameters but induce the same output) and pick a parameter that is symmetric to the original parameter but has “better” curvatures. Here, we define a “good curvature” in terms of eqs. 7 and 8. These curvatures, then, when used in a Laplace approximation, yield better uncertainty estimates. We have added a discussion about this in Sec. 3.2.
>
>
> References:
> 1. Hendrycks, et al. "Deep anomaly detection with outlier exposure." ICLR 2018.
> 2. Meinke and Hein. "Towards neural networks that provably know when they don't know." ICLR 2020.
> 3. Lee, et al. "Training confidence-calibrated classifiers for detecting out-of-distribution samples." ICLR 2018.
> 4. Kristiadi, et al. "Being Bayesian, Even Just a Bit, Fixes Overconfidence in ReLU Networks." ICML 2020.

---

### Official Review · AnonReviewer3 · 2020-10-27
**An Interesting Predictive Variance Module in form of Laplacian approximation**

**Rating:** 4
**Confidence:** 4

**Review:**

#################
Post-Rebuttal Reviews: Thank the authors for the detailed responses. The proposed approach is an interesting add-up for the Laplacian approximations. However, I think the paper still deserves more works. As far as I am concerned, applying the approach to multi-layers instead of only the output-layer is important. I will keep my score for now.

##################

Bayesian deep learning aims to bring uncertainty quantification to modern neural network models. Laplacian approximation, which transforms a trained deterministic neural network into stochastic by a second-order Taylor approximation, is especially appealing due to its pos-hoc nature. However, since the network parameters are trained and fixed, few parameters can be optimized in Laplacian approximation. To enhance the capacity of Laplacian approximation, this paper proposes the *learnable uncertainty units (LULA)*. The LULA units do not affect the network prediction, but the Hessians of the LULA units are nonzero. In consequence, training LULA units increases the capacity of Laplacian approximation by adapting the Hessians.

In general I think the paper proposes an interesting module for parameterizing the predictive variances. The proposed $v(x)$ in Eq(5) is similar to the expression of the neural tangent kernel (Jacot et. al., ‎2018), thus might be suitable for parameterizing the predictive variance. However, the current version of the paper leaves me several questions regarding the methods.

1. It seems to me that the weight matrix does not affect the prediction as long as it is in the form of $\tilde{W}^{(l)} = [ W_{map}^{(l)} \; 0 \;  \backslash \backslash  \; \hat{W}_1^{(l)} \; \hat{W}_2^{(l)} ]$. However, the proposed method sets $\hat{W}_2^{(l)}$ as zero. Setting $\hat{W}_2^{(l)}=0$ blocks the backpropagation through the augmented hidden units, which does not seem reasonable to me.
2. The approximate Hessian $\hat{\Sigma}$ depends on the augmented weights, do you backprop though it when computing the gradients ?
3. The predictive variance is $J^\top \Sigma J$. For multi-output networks like in classification, how do you compute it efficiently ?

**Experiments**
This paper conducts experiments for out-of-distribution detection in order to test the performance. However, in my perspective, the experimental results are not strong enough to prove its superiority.
1. More experiments on uncertainty quantification can be conducted to support the proposed methods. For example, the adversarial example experiment (Ritter et al, 2018) and the calibration experiment (Guo et. al., 2017). These experiments are only my suggestions. I think any experiment regarding to uncertainty quantification would be helpful.
2. The paper uses Laplacian approximation only for the last layer, and all previous layers act as feature extractor. In this scenario, the proposed method is not that different with a parameterized variance neural network $f_{var}(x)$. Then the Laplacian approximation formulation in the paper is not very useful. Moreover, using only the last layer is not a fair comparison with KFL (Ritter et. al. 2018) either.
3. Compared with KFL+LL and KFL+OOD, the vanilla KFL seems to be better based on Table 1 and Table 2. However, KFL+LL and KFL+OOD are fine-tuned versions of KFL, it is weird that tuning the prior variance hurts the performance.

**A few typos**
Eq(2) misses the const $\log p(D)$;
two paragraphs before prop1, "where d it the resulting number"

**References**
Jacot et. al., ‎2018, Neural Tangent Kernel: Convergence and Generalization in Neural Networks
Ritter et. al. 2018, A scalable laplace approximation for neural networks
Guo et. al., 2017, On Calibration of Modern Neural Networks

---

> ### Author Response · Authors · 2020-11-17
> **Response to AnonReviewer3**
>
> Thank you for the feedback. We would like to address your concerns and questions below.
>
> * **Backpropagation through the augmented hidden units is blocked?** Thank you for bringing this up. Indeed, for lower layers (other than the last layer) $\widehat{W}$ has gradient zero. But this is by design: The purpose of lower layers’ augmented weights are to provide a priori flexibility---if one would like to make the curvature of the augmented loss landscape behave in a certain way, it can be done by setting these lower layers’ $\widehat{W}$’s. In practice, when we do not use last-layer LULA, we set these weights randomly. We found---as Prop. 2 shows---that a priori, these weights could already improve Laplace approximations’ uncertainty estimates. Fine-tuning the last-layer LULA’s weights using our objective improves them further. We have clarified this in Sec. 3.2 and provided an empirical evidence in Appendix C.1.
>
> * **Backpropagation through the Hessian?** Please refer to our answer to AnonReviewer1, point 1.
>
> * **Computing predictive variance?** Indeed in general the computation of the Jacobian is infeasible. However, as we have stated in Sec. 5.1., we used Monte Carlo estimation in our experiments. We only used linearized predictive distribution in our theoretical analysis Prop. 2.
>
> * **More experiments?** We have added calibration results in Tabs. 9 and 11 (the first row of each in-distribution data). Furthermore, we have added results from dataset shift experiments (CIFAR-10-C) [2], in Tab. 3. Please refer also to our response regarding this experiment to AnonReviewer1.
>
> * **Only last-layer Laplace? Difference to networks with variance-output?** While our uncertainty tuning is done in the last-layer of a network, one can apply all-layer Laplace on a LULA-augmented network. We show the results of these all-layer Laplaces in our toy examples (Fig. 1) and UCI regression experiments (Appendix C.2). Finally, LULA with last-layer Laplace (LLL) differs from networks with variance-output in that it is based on a Bayesian approximation [1], and so it quantifies _epistemic_ uncertainty while networks with variance output quantifies _aleatoric_ uncertainty.
>
> * **KFL+LL and KFL+OOD are worse than KFL?** Thank you for noticing this. Indeed this seems to be the behavior of these tuning methods in smaller networks. We have added results for larger networks (20-layer CNN and DenseNet-121) in the Tabs. 9-13 in the appendix. We observed there that KFL+LL and KFL+OOD tend to be better than the vanilla KFL. However, our conclusion stays the same: KFL+LULA achieve the best results among these tuning methods.
>
>
> References:
> 1. Kristiadi, et al. "Being Bayesian, Even Just a Bit, Fixes Overconfidence in ReLU Networks." ICML 2020.
> 2.  Ovadia et al. “Can You Trust Your Model's Uncertainty? Evaluating Predictive Uncertainty Under Dataset Shift.” In Neurips 2019.

---

> > ### Comment · AnonReviewer3 · 2020-11-20
> > **Response**
> >
> > Thanks for your responses.
> >
> > **Backpropagation through the augmented hidden units is blocked?**
> > Sorry I do not think your explanation is strong enough to support your design about setting $\hat{W}_2 = 0$. Training $\hat{W}_2 \neq 0$ as well further increases the prior flexibility, so I don't see a reason why don't do that.
> >
> > **Only last-layer Laplace? Difference to networks with variance-output?**
> > I agree that "one can apply all-layer Laplace on a LULA-augmented network". In fact, I think applying all-layer Laplace (LLL) is much more interesting than conducting only the last-layer Laplace. Therefore, I think LLL should be studied as a major part.

---

> > > ### Author Response · Authors · 2020-11-21
> > > **Thank you very much for the swift response.**
> > >
> > > Thank you very much for the swift response.
> > >
> > > For your first point: Indeed training $\widehat{W}_2$, i.e. not forcing it to zero, would further increase the prior flexibility. However, crucially the main point of this paper is to _only_ train the uncertainty, without affecting the function that the MAP-trained network represents. The constraint $\widehat{W}_2 = 0$ is a key factor for ensuring this. Please refer to Prop. 1 (and its proof in the appendix).
> > >
> > > Regarding all-layer Laplace: This issue seems to be nearly orthogonal to our paper in our view. Previous works have studied and compared last-layer (LLL) against all-layer Laplaces [1] (or more generally last-layer against all-layer Bayesian methods [2, 3, 4, etc.]) and concluded last-layer Bayesian methods are competitive to the all-layer counterparts, even though the latter is much more expensive. Thus, LLL is useful for cheaply applying Laplace approximations to very deep networks (e.g. DenseNet-121 in the appendix). Finally, by showing results on LLL, we essentially show that LULA can improve the UQ performance of even such a simple Bayesian method, making it much more competitive to more expensive, strong baselines such as Deep Ensemble.
> > >
> > > Refs.
> > > 1. Kristiadi, et al. "Being Bayesian, Even Just a Bit, Fixes Overconfidence in ReLU Networks." ICML 2020.
> > > 2. Ovadia et al. “Can You Trust Your Model's Uncertainty? Evaluating Predictive Uncertainty Under Dataset Shift.” In Neurips 2019.
> > > 3. Brosse, et al. "On Last-Layer Algorithms for Classification: Decoupling Representation from Uncertainty Estimation." arXiv 2020.
> > > 4. Ober & Rasmussen. "Benchmarking the Neural Linear Model for Regression." arXiv 2019.

---

> > > > ### Comment · AnonReviewer3 · 2020-11-22
> > > > **Reponse**
> > > >
> > > > Thank you for your responses.
> > > >
> > > > For the first point, my point is that even if you set $\hat{W}_2=0$, the function still equals to the original function (**Am I wrong about this ?**). Therefore, we can add this freely in your model without any compromises. I think adding it will make the resulting model much more expressive, given that it passes more gradient information.

---

> > > > > ### Author Response · Authors · 2020-11-23
> > > > > **Thanks a lot for your prompt response**
> > > > >
> > > > > Thanks a lot for your prompt response.
> > > > >
> > > > > Yes, you are right, we can maintain the original network function even if we remove the constraint $\widehat{W}_2^{(l)} = 0$, as long as the last-layer LULA weight matrix deactivates the penultimate LULA units. Thank you for this very useful suggestion, which we will add to the text (and add a thankful note in the acknowledgments)!
> > > > >
> > > > > We have applied a revision to the manuscript to reflect your suggestion, which consists of minor changes in eq. 6 and the proof of Prop. 1.
> > > > >
> > > > > (For clarity, especially to other readers: Since we focus on last-layer LULA, mostly due to computational constraints, this generalization does not affect our experimental results).

---

### Official Review · AnonReviewer1 · 2020-10-28
**Interesting idea with some unclear points**

**Rating:** 6
**Confidence:** 4

**Review:**

***
POST DISCUSSION UPDATE
***
I am satisfied with the authors' response and will increase my score to an accept.
***
END OF UPDATE
***

The paper proposes an uncertainty estimation method for deep learning. The idea is, building on prior work on Laplace approximations for neural networks, to augment a pre-trained network with additional units such that predictions with point estimates remain unaffected, while the variance may change. The weights of those units are then trained on the validation set/out-of-distribution (OOD) data such as to minimise the variance on the in-distribution data and maximise it on OOD data.

Overall, I think this is a novel and interesting method, which could be useful in practice considering that it builds on pre-trained networks. There are a couple of points which are unclear to me (see below for details) as well as some details regarding the experiments which could be improved, so that I would **not recommend acceptance** at this point, however I believe that these can be rectified over the discussion period.

Unclear points:
* How exactly are you training the LULA weights considering that the Hessian (and hence the variance of the corresponding weights and therefore the samples in the MC integral in eq 8) depends on them? I assume you are treating the Hessian as constant? Surely differentiating through its computation would be prohibitively expensive considering that it requires a pass over the entire training dataset? Would it be possible to construct an example (e.g. a reduced version of MNIST) where you can differentiate through the Hessian to see if this has much of an impact?
* I'm somewhat confused by the discussion regarding the closed-form approximations for the predictive distribution around eq. 4. Isn't the point of those to avoid an MC approximation of the integral over the posterior? I understand that computing eq. 5 efficiently for a Laplace approximation over all weights is not generally supported in Pytorch and tensorflow, but wouldn't it be feasible for the last-layer approximation that you use? Or is the whole discussion mainly to motivate the LULA objective in eq. 7?
* Are you using the multi-class equivalents of eq 4 and 5 for test-time prediction or do you do a MC integral over the approximate posterior?

Experiments:
* The setup seems slightly unfair to me in that proposed method is the only one (except KFL+OOD) to be tuned on out-of-distribution data. It would be good to see the DPNs included in Tab. 1 & 2.
* In Tab. 1 it looks like the test predictions from the proposed method are somewhat underconfident. Could you also report test log likelihoods?
* That being said, I find the evaluation a bit narrow in that it only looks at out-of-distribution detection. Could you add e.g. an evaluation of the robustness to data shift? For example, making the predictions using your trained networks for the corrupted CIFAR10 and CIFAR100 variants from (Hendrycks & Dietterich, 2019) shouldn't be too much trouble. See e.g. (Ovadia et al., 2019) for a recent use of those datasets in the literature. I understand that if you're making predictions using eq. 4 accuracies would be unaffected, but it would still be interesting to see log likelihoods and calibration errors.
* Would it be possible to use a more recent architecture for CIFAR100? 50% test accuracy seem somewhat subpar e.g. compared to even small Resnets (which get over 70%).
* Finally, as more of an optional request/suggestion, it would be nice to have some experiment in a domain other than image classification. Considering that you're only implementing inference for the last layer, I would imagine that your method would be compatible with any architecture (e.g. RNN, Transformer) that feeds into a fully-connected output layer? Having more variety in the empirical tasks would strengthen the paper.

References:
Hendrycks & Dietterich. Benchmarking Neural Network Robustness to Common Corruptions and Perturbations. In ICLR 2019.
Ovadia et al. Can You Trust Your Model's Uncertainty? Evaluating Predictive Uncertainty Under Dataset Shift. In Neurips 2019.

---

> ### Author Response · Authors · 2020-11-17
> **Response to AnonReviewer1**
>
> Thank you for your feedback. We will address your concerns and questions below.
>
> * **How exactly are you training the LULA weights considering that the Hessian depends on them?** In practice, we use a stochastic version of Alg. 1: For each minibatch, we compute the last-layer Hessian (we use diagonal Fisher approximation) and treat it as constant, as you have mentioned, to compute LULA’s loss.
>
> * **Surely differentiating through its computation would be prohibitively expensive considering that it requires a pass over the entire training dataset?** By using the stochastic training above, a pass over the entire training set is not required.
>
> * **Would it be possible to construct an example where you can differentiate through the Hessian to see if this has much of an impact?** In our stochastic training, backpropagation through the last-layer Hessian poses no problem for any dataset. (The Fisher approximation of the Hessian only depends on the gradients over the minibatch and modern frameworks---e.g. PyTorch---support double backprop.)
>
> * **Discussion re. eq. 4? Isn't its point to avoid MC-integral?** Indeed it is arguably a better alternative to MC-integrated predictive distribution [1, 2]. However, in this paper, we mainly use it for our theoretical analysis (Prop. 2).
>
> * **Eq. 4 is feasible for the last-layer approximation? Or is it to motivate the LULA objective in eq. 7?** We use the linearization used in eq. 4 to obtain results in Figure 1 and UCI regression in Appendix C.2. But you are right that it can alternatively be used to compute each summand in the first line of eq. 8 in the case of general last-layer training.
>
> * **Are you using the multi-class equivalents of eq 4 and 5 for test-time prediction?** We use Monte Carlo estimation during test-time.
>
> * **DPN in Tabs. 1 and 2?** We show the OOD-detection comparison a la Tabs. 1 and 2 between KFL+LULA and DPN on MNIST in Tab. 7 in the appendix. We found that KFL+LULA outperformed DPN. Unfortunately, after many tries, we were unable to train DPN on larger datasets and networks: We keep getting validation accuracy of ~10% and ~1% for CIFAR-10, SVHN and CIFAR-100, respectively, throughout the training process even when using the hyperparameter values suggested in the DPN’s paper.
>
> * **Underconfident results?** Considering the accuracies of the networks, KFL+LULA is not grossly underconfident, e.g. 88.8% and 44.4% MMCs on CIFAR-10 and CIFAR-100 are close to their accuracies of 90% and 50%, respectively. This indicates LULA does not destroy models’ calibrations. To give empirical evidence: We have added calibration results, in terms of ECE, in Table 9 and 10 (the first line of each in-distribution dataset) in the appendix---LULA yield competitive calibration performance.
>
> * **Dataset shift experiments?** Thank you for the suggestion. We have added results on the dataset shift experiment in Table 3 (Sec. 5.1) as suggested. From these results we observed that LULA provides the best performance compared to other tuning methods for Laplace approximations. Furthermore, we found that it outperformed Deep Ensemble in both calibration error and log-likelihood.
>
> * **Would it be possible to use a more recent architecture for CIFAR100?** Sure, we have added additional results using larger networks---20-layer self-normalizing network and DenseNet-121---in Table 9-13 in the appendix. These results do not change our conclusion that LULA is effective and efficient for improving the UQ performance of these networks.
>
> * **Experiments on other domains?** The results for non-image experiments, in the form of UCI regressions, are presented in Appendix C.2. We agree that our method and last-layer Laplace in general can be applied to broader range domains. However, for the context of this paper, we believe both UCI regression and image classification have already shown the competitiveness of LULA.
>
>
> Refs.
> 1. Foong, et al. "In-Between Uncertainty in Bayesian Neural Networks." arXiv 2019.
> 2. Immer, et al. "Improving predictions of Bayesian neural networks via local linearization." arXiv 2020.

---

### Official Review · AnonReviewer4 · 2020-10-29
**Simple, yet powerful idea to tune uncertainty.**

**Rating:** 7
**Confidence:** 3

**Review:**

POST DISCUSSION UPDATE
------------------------------------
I like the proposed method and I will keep my score.

END OF UPDATE
------------------------------------
The paper proposes a new method to learn uncertainty under Laplace approximations. The method relies on uncertainty units that do not change the predictions but increase the dimensionality of the parameters and help learn better uncertainty by the proposed loss function.

Strong points:

- The paper proposes an ad-hoc method that can be applied to any MAP trained network.
- The proposed method is simple, powerful, and not expensive relative to the training time.
- Good experimental analysis.


Clearly state your recommendation (accept or reject) with one or two key reasons for this choice.
- I recommend to accept this paper.
- This type of work is definitely needed to enable more powerful models that have accurate predictions and better uncertainty estimates with small additional cost.


Questions:
- What are the hyperparameters that the method is sensitive for? For example, the size of OOD, number of epochs, learning rate,..,etc
- How was the number of LULA units chosen? Why did you choose 64? Why not try 32 or less?
- Were the hyperparametrs of LULA tuned on a separate validation set than the one used to tune the uncertainty?

Minor comments:
- Table 1 is very hard to read. It would be great if the best numbers are in bold as in Table 2.

---

> ### Author Response · Authors · 2020-11-17
> **Response to AnonReviewer4**
>
> Thank you very much for your supportive review. We have made the best numbers in Table 1 bold as suggested. We would like to address your questions below.
>
> * **What are the hyperparameters that the method is sensitive for?** The number of LULA units is the primary choice affecting uncertainty quantification performance. Larger number of LULA units tends to make the network underconfident. We present the sensitivity analysis in Sec. C.3 and Table 6 in the appendix.
>
> * **How was the number of LULA units chosen?** We use grid search over the set $\\{ 64, 128, 256, 512 \\}$ to pick the number that balances the in- and out-distribution MMCs (we use 64 units for our experiments in the main text). We have added the detail in Appendix B, in particular eq. 9. We have also added smaller numbers of LULA units (16, 32) in the sensitivity analysis in Table 6. We found that around 32 and 64 units is optimal.
>
> * **Separate validation set for hyperparameter search?** We use the training set to tune LULA’s uncertainty (i.e. to carry out Alg. 1). We have corrected the sentence just below eq. 8: “... we can simply set $\mathcal{D}_\text{in}$ to be the ~~validation~~ training set ...”. To pick the hyperparameter, i.e. the number of LULA units, we use the validation set (see the newly added Appendix B and also Appendix. C.3).

---

### Decision · Program_Chairs · 2021-01-07
**Final Decision**

**Decision:**

Reject

**Comment:**

The paper proposes a simple modification to Laplace approximation  to improve the quality of uncertainty estimates in neural networks.

The key idea is to add “uncertainty units” which do not affect the predictions but change the Hessian of the loss landscape, thereby improving the quality of uncertainty estimates. The “uncertainty units” are themselves trained by minimizing a non-Bayesian objective that minimizes variance on in-distribution data and maximizes variance on known out-of-distribution data. Unlike previous work on outlier exposure and prior networks, the known out-of-distribution data is used only post-hoc.

While the idea is interesting and intriguing, the reviewers felt that the current version of the paper falls a bit short of the acceptance threshold (see detailed comments by R3 and R2’s concerns about Bayesian justification for this idea). I encourage the authors to revise and resubmit to a different venue.